# Effective dynamics of nucleosome configurations at the yeast *PHO5* promoter

Michael Roland Wolff[1], Andrea Schmid[2], Philipp Korber[2]*, Ulrich Gerland[1]*

[1]Department of Physics, Technical University of Munich, Garching, Germany; [2]Molecular Biology Division, Biomedical Center, Faculty of Medicine, Ludwig-Maximilians-Universität München, Planegg-Martinsried, Germany

**Abstract** Chromatin dynamics are mediated by remodeling enzymes and play crucial roles in gene regulation, as established in a paradigmatic model, the *Saccharomyces cerevisiae PHO5* promoter. However, effective nucleosome dynamics, that is, trajectories of promoter nucleosome configurations, remain elusive. Here, we infer such dynamics from the integration of published single-molecule data capturing multi-nucleosome configurations for repressed to fully active *PHO5* promoter states with other existing histone turnover and new chromatin accessibility data. We devised and systematically investigated a new class of 'regulated on-off-slide' models simulating global and local nucleosome (dis)assembly and sliding. Only seven of 68,145 models agreed well with all data. All seven models involve sliding and the known central role of the N-2 nucleosome, but regulate promoter state transitions by modulating just one assembly rather than disassembly process. This is consistent with but challenges common interpretations of previous observations at the *PHO5* promoter and suggests chromatin opening by binding competition.

**\*For correspondence:**
pkorber@lmu.de (PK);
gerland@tum.de (UG)

**Competing interests:** The authors declare that no competing interests exist.

## Introduction

Eukaryotic DNA is packaged inside the nucleus with several layers of compaction. The most basic layer consists of nucleosomes, where 147 bp of DNA are wrapped around a histone protein octamer (*Luger et al., 1997*). Nucleosomes occupy most of the genome and hinder binding of other proteins to DNA, for example transcription factors (*Venter et al., 1994*; *Bell et al., 2011*; *Lai and Pugh, 2017*), replication machinery (*Chang et al., 2016*), and DNA repair enzymes (*Hauer and Gasser, 2017*). This hindrance can be overcome by chromatin remodeling enzymes. Such 'remodelers' bind to nucleosomes and convert the energy from ATP hydrolysis into sliding, ejection, (re-)assembly or restructuring of nucleosomes (*Bartholomew, 2014*; *Zhou et al., 2016*; *Clapier et al., 2017*). Hence, nucleosomes are not static but are constantly replaced to varying degrees throughout the genome by remodeling processes, for example, in the context of transcription (*Dion et al., 2007*). To decipher the resulting nucleosome dynamics, detailed measurements of both, steady state and dynamic quantities are key.

Nucleosome occupancy, a steady-state quantity, is measured in a cell-averaged way with different techniques (reviewed in *Lieleg et al., 2015*). For example, non-nucleosomal DNA is digested with an endonuclease, like MNase, followed by high-throughput sequencing of the nucleosome-protected DNA. Or, DNA is cleaved close to the nucleosome dyad (=midpoint) position by hydroxyl radicals generated in situ at cysteine residues artificially introduced into histones, and the resulting DNA fragments are sequenced (*Brogaard et al., 2012*). After paired-end sequencing, the latter method also gives the distances between neighboring nucleosomes on the same DNA molecule. While such data sets provide quite reliable information on population averaged individual nucleosome positions and nucleosome occupancy, they have two disadvantages: First, the nucleosome occupancy in

absolute terms, that is, the fraction of molecules where a certain position falls within a nucleosome, cannot be determined as only nucleosomal DNA was scored and non-nucleosomal DNA lost from the analysis (for detailed discussion and genome-wide solution to this question see *Oberbeckmann et al., 2019*). Second, the information on the interactions or coupling between several nucleosomes along the same DNA molecule in genomic regions of interest is lost, that is, the obtained occupancy map corresponds only to the one-particle density.

These two disadvantages prevent a more detailed modeling of nucleosome dynamics. In this study, we use the *PHO5* promoter in *Saccharomyces cerevisiae* for which, as a unique exception, single-molecule data for configurations of several adjacent nucleosomes is available. This overcomes both above disadvantages and enables our detailed nucleosome dynamics modeling approach, which integrates steady state and dynamic quantities from four different data sets.

The *PHO5* promoter is a classical and extremely well-studied model system for the role of nucleosome remodeling during promoter activation (reviewed in *Korber and Barbaric, 2014*). It also serves as a paradigm for human promoters. The *PHO5* gene is regulated via the intracellular availability of inorganic phosphate. If phosphate is amply provided, the *PHO5* gene is lowly expressed as its promoter region is occupied by four well-positioned nucleosomes numbered N-1 to N-4 relative to the gene start. Especially nucleosomes N-1 and N-2 occlude transcription factor binding sites that are crucial for gene induction. N-1 occupies the core promoter and prevents access of the TATA-box binding protein (TBP) to the TATA-box. N-2 hinders the transactivator Pho4 from binding its cognate UASp2 element (Upstream Activating Sequence phosphate regulated 2), while the UASp1 element is constitutively accessible in-between N-3 and N-2 . Upon phosphate depletion, a signaling cascade leads to Pho4 activation by inhibition of its phosphorylation and by increasing its nuclear localization (*O'Neill et al., 1996*; *Komeili and O'Shea, 1999*). Pho4 triggers an intricate nucleosome remodeling process involving up to five different nucleosome remodeling enzymes, some with redundant and some with crucial roles (*Musladin et al., 2014*). It also involves histone acetylation, histone chaperones and probably other cofactors that in the end lead to more or less complete removal of nucleosomes N-1 to N-5 and transcription of the *PHO5* gene (reviewed in *Korber and Barbaric, 2014*). Importantly, this chromatin transition was historically among the first shown to be a prerequisite and not consequence of transcription initiation (*Almer and Hörz, 1986*; *Fascher et al., 1993*; *Venter et al., 1994*). Thus, it strongly argued for the now widely accepted view that chromatin structure, positioned nucleosomes in particular, are not just scaffolds for packaging DNA and passive structures during genomic processes, but constitute an important level of regulation.

While the involved cofactors for this model system are known to an exceptional degree, it still remains to be understood which kind of nucleosome dynamics these cofactors bring about. To elucidate these, the full promoter nucleosome configurations in different states are needed. Indeed, for the *PHO5* promoter this information was derived from an electron microscopy (EM) single-molecule scoring approach, at least for the subsystem of the N-1, N-2 and N-3 positions, in several states (activated, weakly activated, and repressed; *Brown et al., 2013*) and from single-molecule DNA methylation footprinting (*Small et al., 2014*). The N-1 to N-3 subsystem of *PHO5* promoter chromatin has been validated before to faithfully recapitulate the regulation of the full-length promoter (*Fascher et al., 1993*). In the single-molecule *PHO5* promoter EM study (*Brown et al., 2013*), simple biologically motivated network Markov models were used to describe the dynamics of promoter nucleosome configurations. As the variation in nucleosome configurations is intrinsically stochastic, that is, is not the result of other stochastic events in the nucleus (*Brown and Boeger, 2014*), the use of Markov models for promoter nucleosome configuration dynamics is valid. However, such models were not systematically investigated but rather several similar models that fit the data of different promoter states presented without further justification or consideration of alternative models. Other, more mechanistically detailed computational models of nucleosome remodeling with base-pair resolution using the data from *Brown et al., 2013*, needed a lot of assumptions to fit the data and therefore do not represent a fully unbiased approach (*Kharerin et al., 2016*).

Since it is yet impossible to observe changes in nucleosome configurations at the same promoter over time in vivo or in vitro, these dynamics have to be inferred by systematic and unbiased theoretical modeling. In theory, each steady state distribution of promoter configurations can be the results of many equilibrium as well as non-equilibrium models. In equilibrium models, the free energy values of each configuration determine the steady state distribution while the reaction rates of each pair of reverting reactions are variable, as long as the ratio of rates corresponds to the Boltzmann factor of

the difference in free energy. This results in zero net fluxes between different configurations in steady state, that is, the average number of reactions in time between any two configurations is equal. In non-equilibrium models, the simple picture of an energy landscape breaks down and net fluxes in steady state are not always zero, which allows for much richer dynamics, for example cycles, trajectories over several configurations with same start and end, which are more likely to occur in one direction than in the other. The challenge in modeling the promoter nucleosome dynamics is to find well-motivated restrictions and assumptions to reduce the number of fitted parameters to a reasonable level, while still staying unbiased, modeling on a similar level as the available experimental data and combining as many different experimental data sets as possible within the same model.

In our study, we achieved this by compiling a large and unbiased collection of possible models, mostly non-equilibrium, within our class of 'regulated on-off-slide models' and selecting the models that are consistent with experimental data. Specifically, we integrate four different experimental data sets: the *PHO5* promoter nucleosome configuration data from *Brown et al., 2013* of repressed, weakly activated, and activated cells, data from our own restriction enzyme accessibility experiments of two different 'sticky (=lower accessibility) N-3' mutant promoters, to address the coupling between remodeling of the N-3 and N-2 nucleosome, and two data sets of Flag-/Myc-tagged histone exchange dynamics experiments (*Dion et al., 2007*; *Rufiange et al., 2007*) to obtain a time scale. Our regulated on-off-slide models include assembly and disassembly of N-1, N-2, and N-3 nucleosomes as well as nucleosome sliding from one occupied position to an unoccupied neighboring position. Additionally, they can mimic regulated transitions from repressed over weakly activated to fully activated promoter state dynamics without changing the network topology and, once fitted, provide likely trajectories of promoter nucleosome configurations, which are completely inaccessible by experiments so far. After systematic analysis of all possible regulated on-off-slide models up to a fixed number of fitted parameters, we found only very few models in agreement with all four data sets and were able to describe the effective dynamics of nucleosome configurations during chromatin remodeling at the *PHO5* promoter in yeast.

## Results

### General modeling approach

Our goal was to (i) create an unbiased class of effective minimal models featuring assembly, disassembly and sliding processes, which describe promoter configurations with the same detail as the available data sets and then (ii) investigate all models within this class to look for the least complex models to explain the data. *PHO5* promoter nucleosome dynamics can be viewed in two different levels of detail. The site-centric point of view focuses on the three positions N-1, N-2, and N-3 (in the following also referred to as N-1, N-2, and N-3 'sites' for modeling purposes) and remodeling at these sites individually, that is, without considering the nucleosome occupancy of neighboring sites (*Figure 1A*). An experimental example are accessibility measurements by restriction enzymes at these positions, since the measurements are taken at each position separately. A more detailed point of view keeps track of all the eight promoter nucleosome configurations ('promoter configurations'), that is, simultaneously tracks which of the three sites are occupied in each cell (*Figure 1B*). We used the configurational data for three cases corresponding to different states of the *PHO5* promoter ('promoter states'): repressed (wild-type), weakly activated (*pho4[85-99] pho80Δtata* mutant), and activated (*pho80Δ* mutant) (*Brown et al., 2013*). Each promoter state shows a different steady state distribution for the occurrences of promoter configurations (*Figure 1C*). It is possible to calculate the three absolute site accessibilities from the promoter configuration occurrences, but important information is lost and the calculation cannot be inverted. To make full use of the information of the data of *Brown et al., 2013*, we based our models on the eight promoter configurations. To further restrict our models, we also used other available site-centric data, like restriction enzyme accessibility at the N-2 and N-3 site for wild-type and two sticky N-3 mutants (this study), as well as published data for histone exchange dynamics at the N-1 and N-2 site (*Dion et al., 2007*; *Rufiange et al., 2007*).

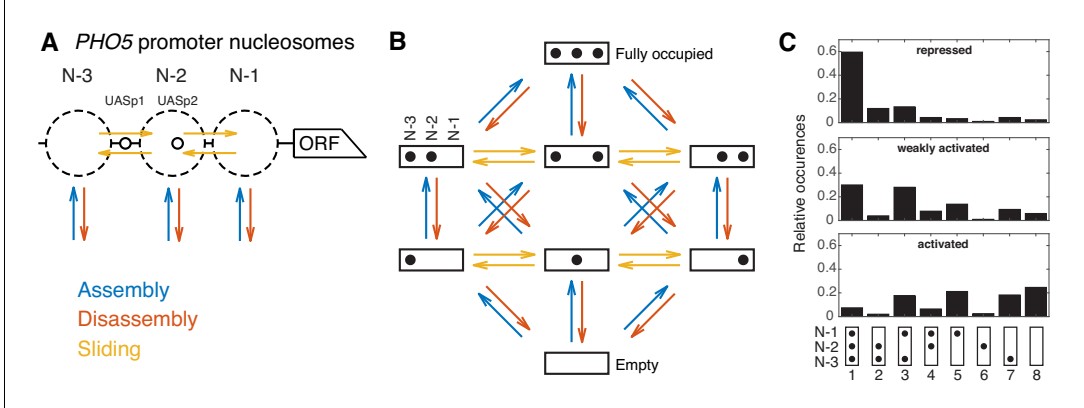

**Figure 1.** Modeling approach and promoter configuration occurrences. (A) Simplified nucleosome dynamics at the *PHO5* promoter including assembly, disassembly and sliding from a site-centric point of view. Dashed circles indicate possible nucleosome positions. Pho4-binding sites (UASp elements) are represented by small circles. (B) Configuration-specific modeling approach with eight promoter configurations and 32 reactions. Arrow color code as in panel A. (C) Measured relative occurrences of the eight promoter configurations indicated at the bottom as in panel B but rotated by 90° and for three different 'promoter states': the repressed wild-type, a weakly activated mutant (*pho4[85-99] pho80Δtata*) and the activated mutant (*pho80Δ*), using data from *Brown et al., 2013*.

The online version of this article includes the following source data for figure 1:

**Source data 1.** Occurrences of promoter configurations measured by *Brown et al., 2013*.

## Regulated on-off-slide models

Each regulated on-off-slide model consists of a set of processes for nucleosome assembly, disassembly and sometimes sliding that allow reactions from one configuration to another (colored arrows in *Figure 1B*). These processes, which will be explained in the following, are the possible building blocks that define each model, that is, determine the transition rate matrix in the Master equation of the underlying Markov process for all three promoter states (Materials and methods). To describe all three promoter states within the same model, at least one of its processes has to be regulated, that is, change its rate to achieve different configuration occurrences and dynamics between different promoter states. Invoking the principle of Occam's razor, we started with global processes and then replaced these in some cases with more specific processes, making the simplest models more and more complex until we found agreement with the considered data. A conceptually similar framework was used to model combinatorial acetylation patterns on histones, but with fixed global disassembly and without the possibilities of sliding and regulation (*Blasi et al., 2016*). Our approach can be applied to any system that features assembly and disassembly reactions on a fixed number of position/sites. The most important input is the steady state distribution of system configurations which can be accompanied by other known steady state or dynamic properties.

### Global processes

The starting point was a model only consisting of two processes, global (i.e. *PHO5* promoter-wide) nucleosome assembly ('A') and disassembly ('D'). Remodeling enzymes can bind nucleosomes and slide them along the DNA (*Bartholomew, 2014*; *Zhou et al., 2016*), so we also included optional sliding processes, which move nucleosomes to a neighboring empty site. This yields another global process, global sliding ('S') (*Figure 2A*).

### Regulation

We define a regulated on-off-sliding model by its set of processes combined with the information which of the processes are regulated. 'Regulated' processes may take on different rate values for different promoter states, whereas 'constitutive' processes have the same rate value for each promoter state. The changing rates of regulated processes are supposed to model the effects caused by transcription factors and/or other factors which influence the promoter state, such as recruited remodeling enzymes, in a coarse-grained fashion. The two simplest models have only the global assembly

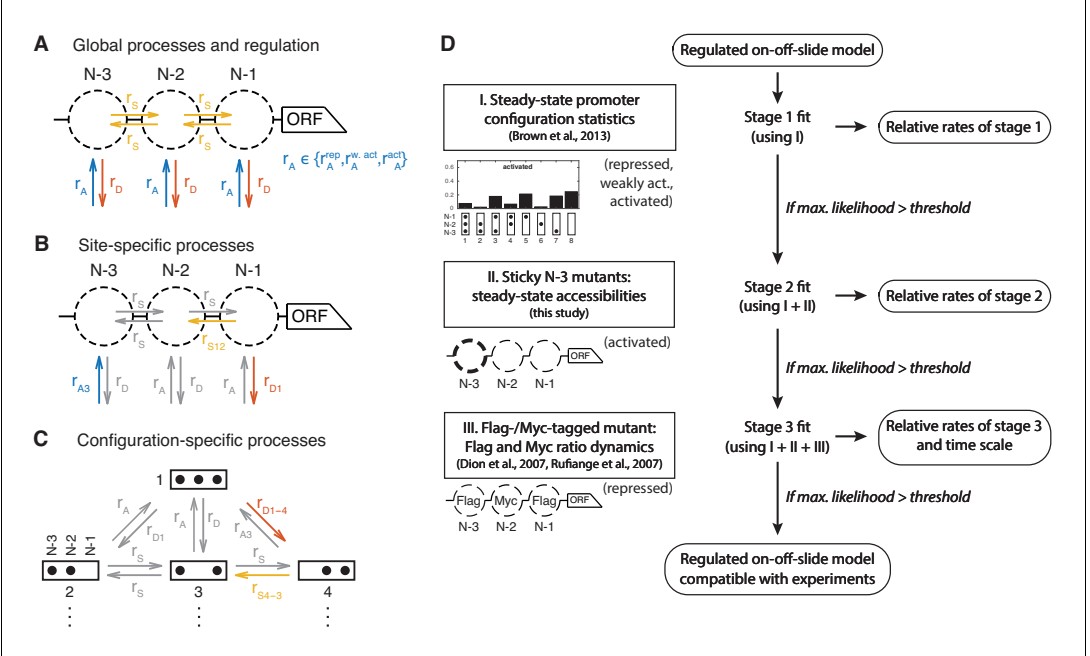

**Figure 2.** Definition and evaluation workflow of regulated on-off-slide models. On-off-slide models consist of processes from different hierarchies: (**A**) Global processes for assembly, disassembly and (optional) sliding. Global processes govern all reactions of the corresponding type with the same rate $r_A$, $r_D$ or $r_S$. To fit multiple promoter states simultaneously, some processes have to be regulated, that is, have different rate values depending on the promoter state. In this example, the global assembly process is regulated. (**B**) Optional site-specific processes for assembly and disassembly at each position (example here with rates $r_{A3}$ and $r_{D1}$) and for sliding between each neighboring pair of positions (here $r_{S12}$). Reactions in gray have not been overruled by more specific processes (here: site-specific processes) and consequently are still determined by the rate parameters of processes on the less specific hierarchy level (here: global processes). (**C**) The last hierarchy level is given by optional configuration-specific processes governing only one reaction (here with rates $r_{D1-4}$ and $r_{S4-3}$). Here, only the promoter configurations 1 to 4 are shown. (**D**) Each regulated on-off-slide model is fitted and evaluated successively using the experimental data on the left-hand side (promoter states during the experiment given in parentheses). Models are discarded if they do not match the maximum likelihood threshold after each stage. With each additional experimental data set, the fit results in new optimal relative rate values of the model. Only the dynamic Flag-/Myc-tagged histone measurements enable us to also fit the time scale.

The online version of this article includes the following figure supplement(s) for figure 2:

**Figure supplement 1.** Occurrences of the different processes in the models with satisfactory likelihood at the different stages.
**Figure supplement 2.** Stage 1: logarithmic likelihood ratio histograms.
**Figure supplement 3.** Stage 1: top 30 models with likelihood above the threshold.
**Figure supplement 4.** Stage 1: models with likelihood above the threshold and only six fitted parameters.
**Figure supplement 5.** Reactions involving the sticky N-3 position.
**Figure supplement 6.** Stage 2: logarithmic likelihood ratio histograms.
**Figure supplement 7.** Stage 2: prefactor histograms.
**Figure supplement 8.** Stage 2: models with likelihood above the threshold.

and global disassembly process, one being regulated and the other being constitutive. If a model is fitted to three different promoter states, one has to fit the three rate values for each regulated process, and one rate value for each constitutive process. Thus regulated processes have three fitted parameters, whereas constitutive processes have one, giving four fitted parameters for the two simplest models. One degree of freedom of the fit corresponds to the overall time scale of each model.

## Site-specific assembly and disassembly processes

*Brown et al., 2013* showed that only global processes are not sufficient to fit the measured occurrences of all eight configurations, even to describe only one promoter state (e.g. the activated promoter). There have to be local modifications of at least one global process (assembly, disassembly or sliding). In *Brown et al., 2013*, modifications were introduced by setting certain reaction rates in the configuration network to zero, effectively searching for a good network topology in the vast discrete space of all possible topologies. In contrast, our approach has the ability to continuously

deviate from the global process rate values for a given set of reactions. Here, the simplest modification is given by site-specific processes (*Figure 2B*). Possible biological mechanisms could be sequence-dependent effects, recruitment or inhibition of remodeling factors in a site-specific way or local differences in the epigenetic marks of nucleosomes. To incorporate such possibilities, we added site-specific assembly, disassembly and sliding processes to the pool of optional processes (see example *Figure 2B*). All processes, except global assembly and global disassembly, are optional and the more optional processes are allowed, the larger the number of all possible models will become. To distinguish the processes at the three nucleosome positions N-1, N-2, and N-3, we named the three site-specific assembly processes 'A1', 'A2', and 'A3', and the three disassembly processes 'D1', 'D2', and 'D3', respectively.

## Site-specific sliding processes

To allow directional sliding, as observed for example in vivo for ISW2 and suggested for RSC (*Whitehouse et al., 2007*; *Krietenstein et al., 2016*), we included five optional site-specific sliding processes. One process governs the rates for all sliding reactions leaving from the N-2 site to account for the possibility of start site-specific non-directional sliding ('S2*'). Two processes enable directional sliding between N-1 and N-2 sites and two directional sliding between N-2 and N-3 sites, with the short name 'Sxy' for sliding from site x to site y. Note that these sliding processes actually are not only dependent on the state of one site, but two sites: the origin and the destination of the sliding process. The origin needs to be occupied and the destination to be empty. This already constitutes a correlation between neighboring sites. We still call these processes 'site-specific', since they do not take into account the full configuration.

## Modulation by more specific processes

Site-specific processes govern a subset of reactions of all reactions of a given type (assembly, disassembly, or sliding) to allow deviations from the global process rate at a given position. To achieve this we invoke the following rule, which is also used for the even more specific processes of the following paragraph: If the reactions governed by any two processes in a given model overlap, as for example the global and any site-specific assembly process, the more specific process, that is, the one governing less reactions, determines the rate values of these reactions, 'overruling' the more general process, which then only governs the left-over, non-overlapping reactions. Thus, a more specific process only overrules a more general process that contains the same reactions. This rule is the most general solution for overlapping processes, allowing an increased as well as a decreased rate value for the reactions of the more specific process, that is, the specific process to be enhanced or inhibited with respect to the general process. If all reactions of a process are overruled by more specific processes in a given model, that model is redundant and will be ignored.

## Configuration-specific processes

To allow even more specific modulations, we also added configuration-specific processes which only govern a single reaction rate and overrule any more general process for this reaction (see example *Figure 2C*). Since there are in total 32 reactions, this gives another 32 optional processes. For instance, the disassembly process from configuration 1 to configuration 4, is denoted with 'D1-4', sliding from configuration 4 to configuration 3 with 'S4-3'.

## Resulting model set

With the simplest regulated on-off-slide models having four parameters, we increased the maximal parameter number up to seven to obtain the first models that simultaneously fit all data sets within the experimental error. This yielded models with one regulated process (having three fitted parameter), and 1–4 constitutive processes, as well as models with two regulated processes and one constitutive processes. Models with zero constitutive processes, that is, only regulated processes, are ignored since they decouple the time scales of the different promoter states and in this case the fit to steady state observables is not harmed by setting any single regulated process to a constitutive process. After taking into account all combinations of processes with up to seven parameters in total we ended up with 68,145 regulated on-off-slide models. Here, we ignored duplicate models with identical transition rate matrices constructed with different processes (e.g. rare models where all

reactions of a process are overruled by other processes, making it identical to the model without the overruled process).

The relative occurrence of individual constitutive as well as regulated processes in this initial model set is the same for all assembly and disassembly processes except global assembly and disassembly and slightly lower for non-global sliding processes where the number of effectively identical, and thus ignored, models is higher (*Figure 2—figure supplement 1A*).

## Staging

We used the models of this new class and a step by step ('staged') fitting procedure to determine which of them are capable to reproduce the four experimental data sets (*Figure 2D*). The benefit of this staging approach was that we could dissect the contributions of the different data sets to the model selection as well as reduce model count in the later, computationally more involved, stages. Each stage also fits again the data from the previous stages. Thus, models that drop out at an earlier stage cannot improve in a later fit to more data when using the same threshold in each stage, making the results independent of the exact staging order. Note that, even in a different order, the configuration statistics are needed in all stages to determine well-defined relative rates for the processes of each model, thus making it part of stage 1.

### Promoter configuration statistics

First, in stage 1, the parameter values, that is, the rate values of the involved processes, of each model were determined by maximizing the likelihood to observe the measured *PHO5* promoter configurations (see Materials and methods, *Figure 1C*). The global assembly parameter (for the activated state) was set to one to fix the time scale at first, which yielded relative rates for the remaining six parameter values. Since we will fit the time scale later as well, we always include it in the number of fitted parameters. As an example, consider the model with the processes and reaction network shown in *Figure 3A*. After the maximum likelihood fit, one can compare the different model nucleosome configuration distributions for the three promoter states with the configuration statistics data. Since also including the data sets discussed below into the fit did not worsen the deviations from the configuration statistics data for this specific model (data not shown), we already show here the final nucleosome configuration distributions (*Figure 3B,C and D*).

For model selection, we used the logarithmic likelihood ratio with respect to the perfect fit, $R_1 = -L_I + L_0$, where $L_I$ is the maximum log10 likelihood of a given model in stage 1 and $L_0$ the highest possible log10 likelihood to obtain the configuration data, corresponding to a perfect fit. The best model achieved $R_1 = 4.02$ (distribution of $R_1$ for all models in *Figure 2—figure supplement 2A*) and our example model in *Figure 3* $R_1 = 4.38$. We set an upper threshold to $R_1$, $R_{max} = 6$, to define models that are in good agreement with the measured configuration statistics. We used the same $R_{max}$ in each stage and this threshold, which denotes a likelihood $\approx 100$ times lower than the current best model, gave enough room for fitting models to additional data later on. At this stage, we found 173 such models, with the top 30 models presented in *Figure 2—figure supplement 3*. Two of these 173 models used only 6 instead of the maximum of 7 fitted parameters (with $R_1 = 5.32$ and 5.88, *Figure 2—figure supplement 2D* and *Figure 2—figure supplement 4*).

For comparison, *Brown et al., 2013* used different network topologies to fit the different promoter states with three parameters values per state (assembly, disassembly and sliding, with assembly set to 1). The corresponding logarithmic likelihood ratio $R_1$ of the combined model of the three states is 4.93, using seven fitted parameters, when also counting the time scale. This reveals the power of our systematic approach, since we found models with higher likelihood as well as models with one parameter less and not much worse likelihood.

## Model heterogeneity

At this stage, the set of good candidate models, that is, models with $R_1$ below the threshold, was still quite heterogeneous, that is, there were models where all sliding reactions have positive rates, and models with no sliding at all (*Figure 2—figure supplement 2C*). In fact, this was already the case for the second best model (*Figure 2—figure supplement 3*). Some models with $R_1$ below the threshold did not use any configuration-specific processes (*Figure 2—figure supplement 2B*). The regulated processes were mostly global assembly, while some models showed regulated global

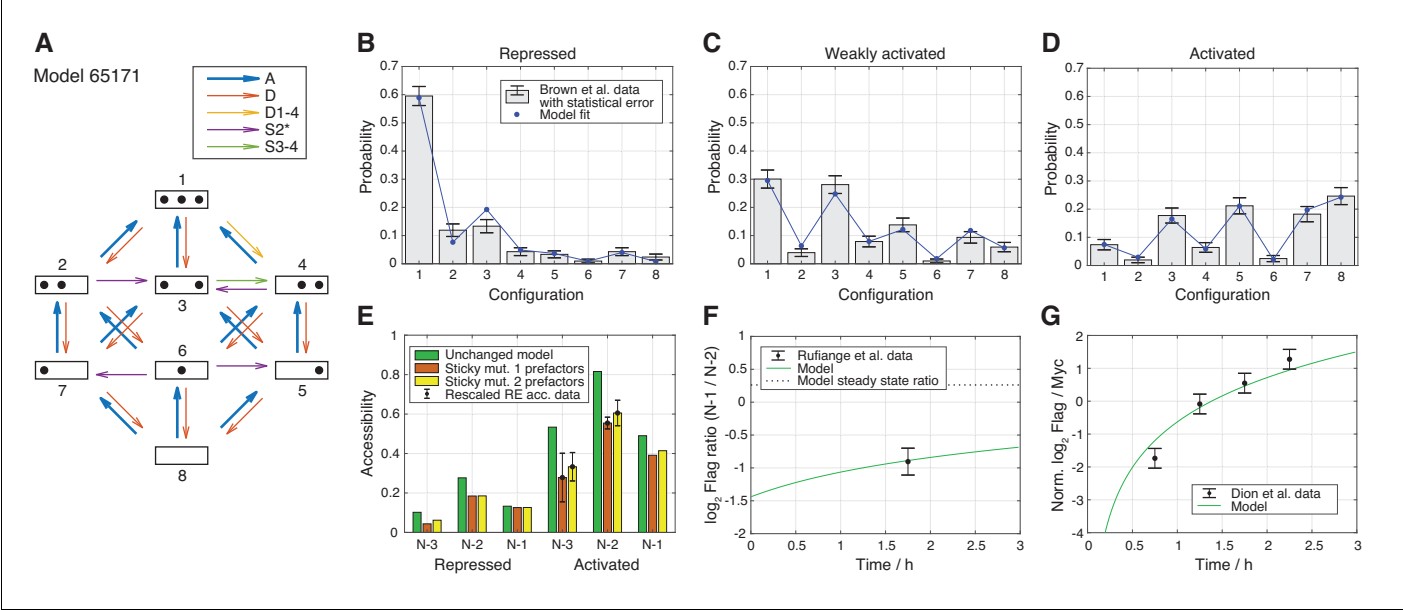

**Figure 3.** Best regulated on-off-slide model compatible with all the experimental data sets in stage 3. All fits were done simultaneously (see Materials and methods). (A) Regulated on-off-slide model with the regulated process A (global assembly, thick arrows) and the constitutive processes D (global disassembly), D1-4 (disassembly from configuration 1 to 4, overrules D), S2* (sliding away from N-2) and S3-4 (sliding from configuration 3 to 4). (B, C, D) Combined fits to the steady-state promoter nucleosome configuration occurrences in repressed, weakly activated, and activated state. Only a change in the rate of the regulated global assembly process accounts for differences in the three distributions. The other processes (D, D1-4, S2*, and S3-4) are constitutive and their rates do not change. The model fits in stage 1 (ignoring all other data) and stage 2 (ignoring Flag-/Myc-tagged histone exchange data) are only slightly better (data not shown). (E) Fit to the sticky N-3 RE accessibility data of two mutants in the activated state (error bars with standard deviation of rescaled RE accessibility). Only reaction rates involving the N-3 were allowed to divert from the previously fitted parameter values. (F) Fit to the *Rufiange et al., 2007* data of Flag amounts at N-1 over N-2 after 2 h of Flag expression (shifted by 0.25 h lag time determined in *Dion et al., 2007*). The error bar corresponds to the standard deviation of two measurements. (G) Fit to the *Dion et al., 2007* data of Flag over Myc amounts at N-1 at four time points after Flag expression (shifted by 0.25 h lag time). y-axis points are normalized by their mean to account for a sloppy fitted parameter in the treatment of the data in *Dion et al., 2007*. Error bars are estimated experimental standard deviations used in the fit.

The online version of this article includes the following figure supplement(s) for figure 3:

**Figure supplement 1.** Stage 3: Flag amount ratios of tested models.

**Figure supplement 2.** Stage 3: Flag over Myc amount ratios of tested models.

**Figure supplement 3.** Stage 3: logarithmic likelihood ratio histograms.

disassembly or site-specific assembly at N-2 (*Figure 2—figure supplement 1B*). In the following stages, we further sorted out models by fitting to additional data sets.

## Assembly-disassembly symmetry only for equilibrium models

Note that, except for a few cases, regulated on-off-slide models are non-equilibrium models. Equilibrium regulated on-off-slide models, that is, models where the net fluxes between all promoter configurations in steady state are zero in all states (also known as 'detailed balance' models), have an assembly-disassembly symmetry. That means, swapping all assembly processes to the equivalent disassembly process and vice versa (e.g. A to D, A1 to D1, D2 to A2), yields a model with equal maximum likelihood. Additionally, for each equilibrium model with a regulated assembly parameter there is a symmetric equilibrium model with the corresponding regulated disassembly parameter and equal maximum likelihood, and vice versa. Non-equilibrium models do not share this symmetry. Among the 68,145 investigated regulated on-off-slide models, there are 196 equilibrium models with zero net fluxes independent of their parameter values. These are models that only use global processes, where the symmetry is trivial, as well as models without any sliding or any configuration-specific processes. These equilibrium models did not fit the data well, with the best $R_1 \approx 11$.

## Integration of *PHO5* promoter mutant data

### Accessibility experiments for *PHO5* promoter mutants

In the next stages, we eliminated the regulated on-off-slide models that did not agree with further data. *Small et al., 2014* published two *PHO5* promoter mutants where the DNA sequence underlying N-3 was mutated such as to increase certain dinucleotide periodicities that favor nucleosome formation and may increase intrinsic nucleosome stability (*Satchwell et al., 1986*). Using an at that time novel DNA methylation assay for probing nucleosome occupancies, these authors published that N-3 at these mutated *PHO5* promoters was hardly removed upon *PHO5* induction. These mutated *PHO5* promoters offered an interesting parameter modulation and we could have used, in principle, these nucleosome occupancy data for further selection among our models. However, for reasons detailed in the Discussion section, we questioned the published data and wished to check them by classical and well-documented restriction enzyme (RE) accessibility assay (*Gregory et al., 1999*). Therefore, it was important that we obtained the exact same strains from *Small et al., 2014* and measured N-2 and N-3 occupancies at the wild type and mutant *PHO5* promoters (*Table 1* and *Table 1—source data 1*). We confirmed that the sticky N-3 mutant promoters show reduced, relative to wild type, removal of both N-2 and N-3 upon *PHO5* induction, but we did not confirm that these nucleosomes were hardly removed at all. Nonetheless, these data now provided additional constraints for our modeling approach as we needed to find the regulated on-off-slide models which show the same interdependence of N-2 and N-3 accessibility.

### Accessibility fold-changes

The *Small et al., 2014* strains included corresponding isogenic wild-type strains and we noted that our RE accessibilities for wild type in the activated state were lower than the accessibilities in the activated state of the wild type calculated from the data from *Brown et al., 2013* (*Table 1*). This discrepancy likely stems from different experimental conditions. The *Brown et al., 2013* study used a different strain background, YS18, and the *pho80* allele in high phosphate conditions for induction, while we used the S288C background and over night phosphate starvation to achieve direct comparison with the data by *Small et al., 2014*. We saw repeatedly that S288C strains do not yield as high ClaI accessibility values in the induced state as the YS18 background, which was formerly used in classical *PHO5* studies reporting such high degree of *PHO5* promoter nucleosome remodeling (*Ertel et al., 2010*). Nonetheless, this discrepancy did not matter for our purposes as we could use the accessibility fold-changes of mutants compared to the wild-type to test our models in order to normalize the accessibility values coming from different experiments.

**Table 1.** Restriction enzyme (RE) accessibility of N-2 and N-3 sites in phosphate starved cells measured in this study and corresponding accessibility values of *Brown et al., 2013* (RE accessibility with mean ± standard deviation of two independent biological replicates and the fold-change standard deviation calculated using standard error propagation).
The sticky N-3 mutants feature manipulated DNA sequences at the N-3 site, which decrease the RE accessibility at the N-3 site compared to the wild-type. In our study, this sticky N-3 also decreases the accessibility of the N-2 site. In stage 2, we tested which regulated on-off-slide models with compatible configuration distribution in stage 1 can at the same time reproduce the accessibility fold-changes at sites N-2 and N-3 for both sticky N-3 mutants.

| | Wild type | Sticky N-3 mutant 1 | | Sticky N-3 mutant 2 | |
| --- | --- | --- | --- | --- | --- |
| | accessibility | accessibility | wt fold-change | accessibility | wt fold-change |
| N-2 RE (ClaI) | 64.0 ± 1.4 % | 43.5 ± 2.1 % | 0.68 ± 0.04 | 47.5 ± 4.9 % | 0.74 ± 0.08 |
| N-2 Brown et al. | 82% | | | | |
| N-3 RE (HhaI) | 58.5 ± 2.1 % | 30.5 ± 13.4 % | 0.52 ± 0.23 | 36.5 ± 7.8 % | 0.62 ± 0.13 |
| N-3 Brown et al. | 55% | | | | |

The online version of this article includes the following source data for  Table 1:
Source data 1. RE accessibility of independent biological replicates in phosphate starved cells.

## Modeling approach

In stage 2, we fitted the experimental fold-changes together with the data used in stage 1, minimizing $R_2 = -(L_I + L_{II}) + L_0$ with $L_{II}$ being the log likelihood to obtain the accessibility fold changes (see Materials and methods). We needed to systematically test for each model whether reasonably large changes in reaction rates involving the N-3 nucleosome could lead to the same behavior seen in the experiment. Since we did not know the exact consequences of the sticky N-3 mutation on the reaction rates or how to translate them within a given model, we had to consider all possibilities of changes in reactions where the N-3 nucleosome is involved. We achieved this by including prefactors for these 12 reaction rates (*Figure 2—figure supplement 5*), which were fitted together with the model parameters to the configuration statistics data and the accessibility fold changes of both sticky N-3 mutants and allowed each prefactor to vary between $1/5$ and $5$ for each sticky N-3 mutant. For equilibrium models, the ratio of a pair of reverting reaction rates can be interpreted by $e^{\Delta E / k_B T}$, with $\Delta E$ denoting the energy difference between the two configurations. This leads to a maximal possible absolute change in N-3 nucleosome binding energy modeled by the prefactors of $|\Delta E_{\mathrm{wildtype}} - \Delta E_{\mathrm{N-3mutant}}| \approx 3.2 \, k_B T \approx 1.9 \, \mathrm{kcal/mol}$. We decided to use four prefactors per sticky N-3 mutant, one for each group of reactions, assembly at N-3, disassembly at N-3, sliding from N-3 to N-2 and from N-2 to N-3 (*Figure 2—figure supplement 5*), respectively.

Using the same likelihood threshold as in stage 1 left us with 15 models that fit both sticky N-3 mutants well (*Figure 2—figure supplement 6A*), with the best logarithmic likelihood ratio of $R_2 = 4.38$ for the model in *Figure 3*, with *Figure 3E* showing a perfect fit to the sticky N-3 fold changes (also including the following data sets). Out of these 15 models, only three models without configuration-specific processes remained (*Figure 2—figure supplement 6B*). With this fit we also excluded models without sliding processes (*Figure 2—figure supplement 6C*) as well as models with less than seven fitted parameters (*Figure 2—figure supplement 6D*).

## Prefactor behavior

The fitted prefactor values show stable qualitative behavior for the models with $R_2$ below $R_{max}$ in both mutants (*Figure 2—figure supplement 7*): consistent with the reduced experimental accessibility at N-3, assembly at N-3 is by a prefactor greater than one while disassembly at N-3 is decreased by a prefactor smaller than 1. Furthermore, in these models the sliding from N-3 to N-2 is increased and the sliding from N-2 to N-3 decreased by the two sliding prefactors. While these favor the accessibility at N-3 again, they lead to the concomitant accessibility decrease at N-2.

## Relative occurrence of sliding

Taking into account the sticky N-3 data has a strong impact on the relative occurrences of sliding processes, as it strongly favors models with either S32 or S3-4 processes, both of which enable the sliding from N-3 to N-2 (*Figure 2—figure supplement 1C*). All 15 good models in stage 2 include at least one sliding process (*Figure 2—figure supplement 8*).

## **Flag-/Myc-tagged nucleosome exchange simulation**

In stage 3, we used histone dynamics measurements to further constrain our models and, for the first time, were able to determine the optimal time scale of each model. Histone dynamics measurements reflect the appearance/disappearance of nucleosomes regardless if via assembly/disassembly or sliding and is measured in cells that constitutively express Myc-tagged histones and are then induced to express also Flag-tagged histones, which, over time, are incorporated into nucleosomes (*Schermer et al., 2005*). We used the histone pool model as well as the measured average Flag over Myc amount ratio at the N-1 in MNase-ChIP-chip assays of *Dion et al., 2007* and the measured Flag at N-1 over Flag at N-2 amount ratio in Flag-tagged H3 MNase-ChIP-qPCR assays of *Rufiange et al., 2007* in the repressed promoter state. We investigated the nucleosome configuration dynamics of each model by keeping track which nucleosome contains a Flag- or Myc-tagged histone H3 (see Materials and methods). This enabled us to calculate the dynamics of the ratio of Flag at N-1 over Flag at N-2 amount (*Figure 3F* and *Figure 3—figure supplement 1A*) and the ratio of Flag over Myc amount at the N-1 (*Figure 3G* and *Figure 3—figure supplement 2*) for each model and determine the agreement with the two histone dynamics data sets. We obtained seven models in agreement with the data, with a best logarithmic likelihood ratio of

$R_3 = -(L_I + L_{II} + L_{III}) + L_0 = 4.61$, with $L_{III}$ being the summed log10 likelihoods of the third and fourth data set (*Figure 3—figure supplement 3A*). Thus, the threshold value of $R_{max} = 6$ denotes a likelihood $\approx 25$ times lower as the best achieved value of $R_3$. We also calculated the dynamics of the Flag amount ratios between the other sites (N-3 over N-2 and N-3 over N-1, see *Figure 3—figure supplement 1B,C*) and similarly the same ratios in the activated promoter state (*Figure 3—figure supplement 1D,E,F*).

## Properties of satisfactory models

### Processes and rate values

Out of the 68,145 models with at most seven fitted parameter values, seven models satisfied our threshold for the likelihood in the combined fit to the nucleosome configuration data, the sticky N-3 mutant accessibility data and the H3 exchange data. These 'satisfactory' models, their processes and rate values are presented in *Figure 4*, with the corresponding reaction rates shown in *Figure 4—figure supplement 1*.

### Only one regulated process

All satisfactory models had one regulated process and four constitutive processes (*Figure 4*). Simultaneous regulation of two processes in combination with one constitutive process (also summing up to seven fitted parameters) did not fit the data well enough, probably because regulation with two processes leaves only one possible constitutive process slot and only three processes in total. Thus,

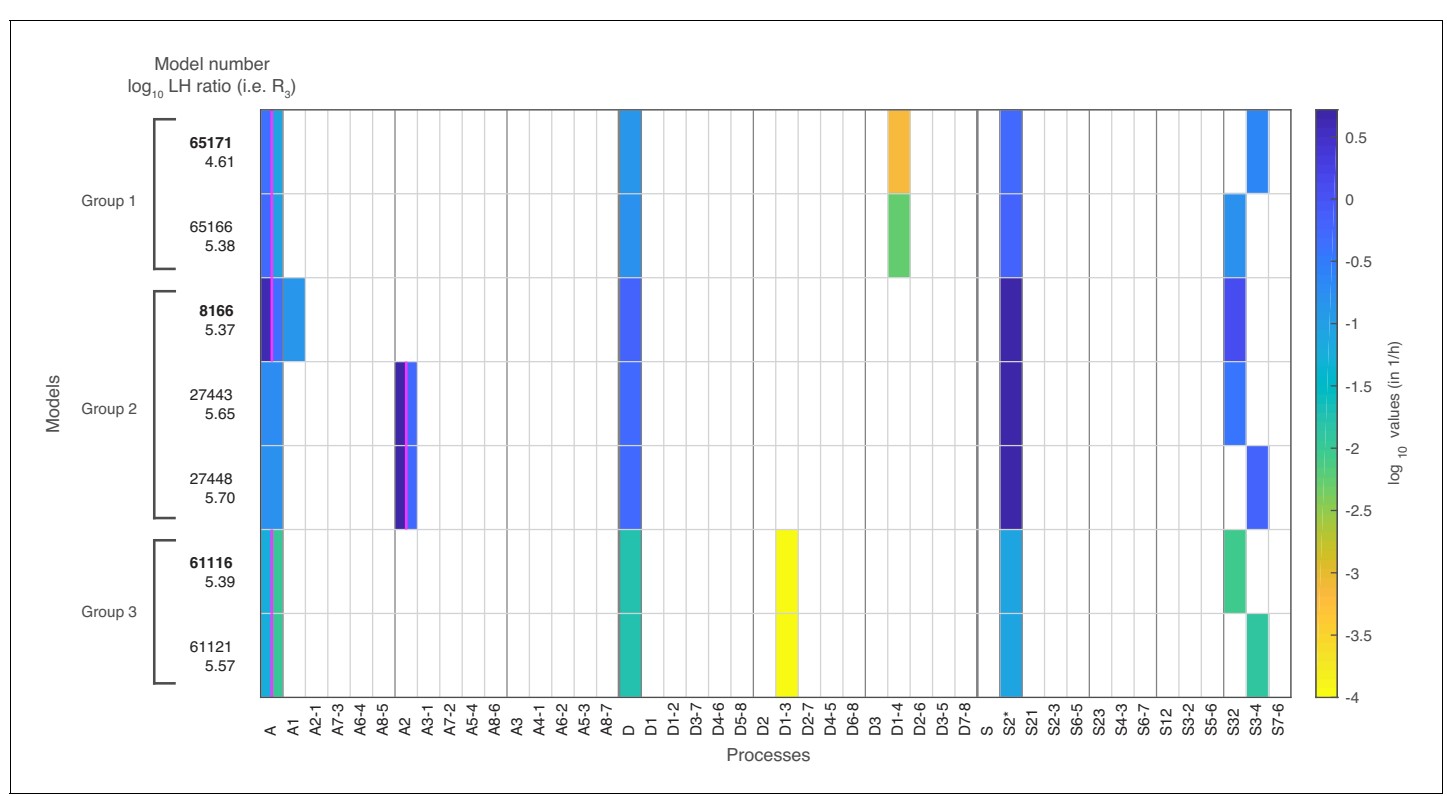

**Figure 4.** The seven models that agree with all four data sets after stage 3. The colored boxes in each row show the model processes and their rate values. White boxes denote the absence of a process in a model. Regulated processes are separated into two differently colored boxes for repressed (left half) and activated (right half) promoter state. Weakly activated rate values are not shown here. On the left side are the model number and the log10 ratio of the best possible likelihood and the model likelihood, $R_3$. The models are grouped with respect to similarities in the site-centric net fluxes (*Figure 5—figure supplement 5*). The groups' representatives are printed in bold with net fluxes compared in *Figure 5*.

The online version of this article includes the following source data and figure supplement(s) for figure 4:

**Source data 1.** Fitted parameter values of constitutive processes and time scale with error estimates.
**Source data 2.** Fitted parameter values of regulated processes with error estimates.
**Figure supplement 1.** Relative rate values for all 32 reactions of the seven satisfactory models.

the rather simple regulation of only one process is preferred over regulation of several processes, when keeping the total number of fitted parameter constantly at 7.

## Regulation by assembly

Within the seven satisfactory models, the regulated processes are global assembly, A (5x) and assembly at N-2, A2 (2x) (*Figure 4* and *Figure 2—figure supplement 1D*). The fact that regulation by assembly rather than disassembly is more likely to reproduce the data can already be observed after the maximum likelihood fit to the configuration data in stage 1 (*Figure 2—figure supplement 1B*). The rates of all regulated assembly processes decrease when going from the repressed over the weakly activated to the activated state. This is in agreement with the total nucleosome occupancy decrease on the promoter.

## Sliding processes needed

Since all sliding processes are optional, we asked whether sliding is needed to fit all data sets. We found that sliding occurs in all seven models, and in each case with at least two different processes, that is, just global sliding with the same rate for each sliding reaction is not sufficient. Every satisfactory model employs the sliding process away from N-2 (S2*) and a process that allows sliding from N-3 to N-2 (*Figure 4* and *Figure 2—figure supplement 1D*).

## Use of site- and configuration-specific processes

On-off-slide models combine processes from different hierarchies. Each model has a global assembly and global disassembly process, but global sliding is optional and not used in any of the satisfactory models. Each of these models has one to three site-specific processes out of A1, A2, S2*, and S32, which overrule global processes. We found two models without configuration-specific processes, models 8166 and 27,443 (*Figure 4*). All other satisfactory models have one or two configuration-specific processes. Thus, configuration-specific processes are helpful, but not needed to achieve agreement with the data. However, that does not mean that the three nucleosome sites can be independent of each other, since the sliding reactions always introduce coupling between neighboring sites.

## Fluxes

Knowing all reaction rates, it is easy to calculate the directional fluxes (*Figure 5—figure supplement 1* and *Figure 5—figure supplement 2*) as well as the net fluxes (*Figure 5—figure supplement 3* and *Figure 5—figure supplement 4*) between all configurations for each satisfactory model and for different promoter states. Since there are no methods available to measure these fluxes experimentally, modeling approaches provide the only view into possible internal promoter configuration dynamics. As stated before, the vast majority of all regulated on-off-slide models are non-equilibrium models, that is, have non-zero net fluxes. In the repressed promoter, the seven satisfactory models showed the highest fluxes as well as net fluxes occurring between the configurations with three or two nucleosomes (*Figure 5—figure supplement 1* and *Figure 5—figure supplement 3*) and cyclic net fluxes from configuration 1 to 2 to 3 and back to 1. Despite qualitative similarities, the fluxes and net fluxes also show the different behavior of the seven models, for example only five of them showed cyclic net fluxes from configuration 1 to 4 to 3 and back to 1. In the activated promoter state, the higher fluxes and net fluxes between the first four configurations are lost and the differences between the seven models become more pronounced.

## Site-centric net fluxes

Going back to the view point of nucleosome sites as in *Figure 1A*, we define effective 'site-centric net fluxes' by summing all assembly/disassembly net fluxes at each site and sliding net fluxes between N-1 and N-2 as well as N-2 and N-3 (*Figure 5—figure supplement 5*). The site-centric net fluxes give a simplified picture of the net paths of the nucleosomes on the promoter, but ignore the events on neighboring sites, which are only correctly depicted in the directional or net fluxes between the promoter configurations. While in all satisfactory models, the N-2 site had a central role with the strongest net nucleosome influx in all promoter states, we also found differences and divided the satisfactory models into three groups. Two models (group 1) have site-centric net influx

only at N-2 and N-3 for the repressed state and only at N-2 for the activated state. The other models have site-centric net influx only at N-2 for all promoter states, but show very different flux amounts. Three models (group 2) show a repressed N-2 site-centric net influx of $\approx 0.59\,\mathrm{h}^{-1}$, while the last two models (group 3) have $\approx 0.013\,\mathrm{h}^{-1}$. The time scale parameters of group 3 are 10-fold lower than for the other two groups and have larger error bars (see Materials and methods and *Figure 4—source data 1*), allowing the time scale to increase by up to a factor of 10 as well as becoming arbitrarily small, while still meeting the fit threshold. Thus, the time scale parameter of this group is not properly determined by the given data. For each group, we picked a representative model (highest likelihood within the group) and show the net fluxes and the site-centric net fluxes in *Figure 5*.

## Maximal reaction rates

After setting the time scale for each model, we investigated the rate values for each reaction (*Figure 4—source data 1* and *Figure 4—source data 2*). Within all satisfactory models, the highest rate of assembly and disassembly processes were $5\,\mathrm{h}^{-1}$ and $0.6\,\mathrm{h}^{-1}$, respectively. Sliding rate values had a maximum of $5\,\mathrm{h}^{-1}$, corresponding to translocation speeds of at least $0.2\,\mathrm{bp/s}$ assuming an unidirectional travel of $160\,\mathrm{bp}$ with constant speed between two sites and instantaneous binding of remodeler enzymes needed for sliding. Experimentally measured translocation speeds of up to $13\,\mathrm{bp/s}$ of yeast SWI/SNF or RSC remodeling complexes in vitro (*Zhang et al., 2006*) leave enough room to account for a delay due to binding of sliding remodelers, possibly making it the limiting factor in the sliding rate of $5\,\mathrm{h}^{-1}$, which would in turn lead to corresponding translocation speeds closer to the experimentally measured value.

## Bounds for chromatin opening and closing times

*PHO5* induction results from consecutive signal transduction, promoter chromatin opening, transcription initiation and downstream processes that lead to functional Pho5 acid phosphatase gene product. On the level of *PHO5* mRNA or acid phosphatase activity, induction starts about two hours after phosphate starvation of the cells (*Rajkumar et al., 2013*; *Barbaric et al., 2001*), while after the same time chromatin opening in terms of chromatin accessibility is usually complete, that is, the kinetics of *PHO5* promoter chromatin opening are faster (*Schmid et al., 1992*; *Barbaric et al., 2001*; *Korber et al., 2006*; *Barbaric et al., 2007*). We modeled here *PHO5* promoter chromatin remodeling and can give approximate upper bounds for the chromatin opening rates for each regulated on-off-slide model ('effective chromatin opening rate', see Materials and methods).

Models in group 1 had an effective chromatin opening rate close to $0.2\,\mathrm{h}^{-1}$. Group 2 showed a faster effective chromatin opening rate of $\approx 0.8\,\mathrm{h}^{-1}$, while group 3 yielded $\approx 0.02\,\mathrm{h}^{-1}$. Since these values are proportional to the time scale, they inherit its uncertainty. The effective chromatin opening rate of group 2 was high enough to reflect the experimentally measured kinetics. The other two groups could also match these kinetics, although just barely, if the maximum time scale uncertainty was considered (approximately an additional factor of 2 for group 1 and a factor of 10 for group 3, see *Figure 4—source data 1*).

Conversely, we did the same calculations for the 'effective chromatin closing rate', which is an upper bound of how fast a given model can switch to the repressed state. The ratio of the effective chromatin closing rate over the opening rate was between 2.9 and 3.5 for all satisfactory models. Even just after the first fit in stage 1, all investigated assembly-regulated models in agreement with the configurational data had a ratio ranging from 1.5 to 4.5. The few disassembly-regulated models after stage 1 had a ratio ranging from 0.25 to 0.65. This further supports the assembly-regulated models as promoter chromatin closing and repression of *PHO5* transcription were experimentally shown to be faster than chromatin opening and transcription activation. After the shift from phosphate-free to phosphate-containing medium, repression of *PHO5* transcription was almost complete within 20 min (*Schermer et al., 2005*) and 65% of chromatin closing was achieved within 45 min (*Schmid et al., 1992*).

We also used our models to directly calculate the dynamics of the nucleosome configuration distributions during chromatin opening and closing. Similar to the calculation of the effective chromatin opening/closing rates, we assumed an instantaneous signal, that is, an abrupt change of the regulated process rates from repressed to activated and vice versa. Interestingly, our models predict a probability maximum after leaving the initial repressed steady state and before arriving in the

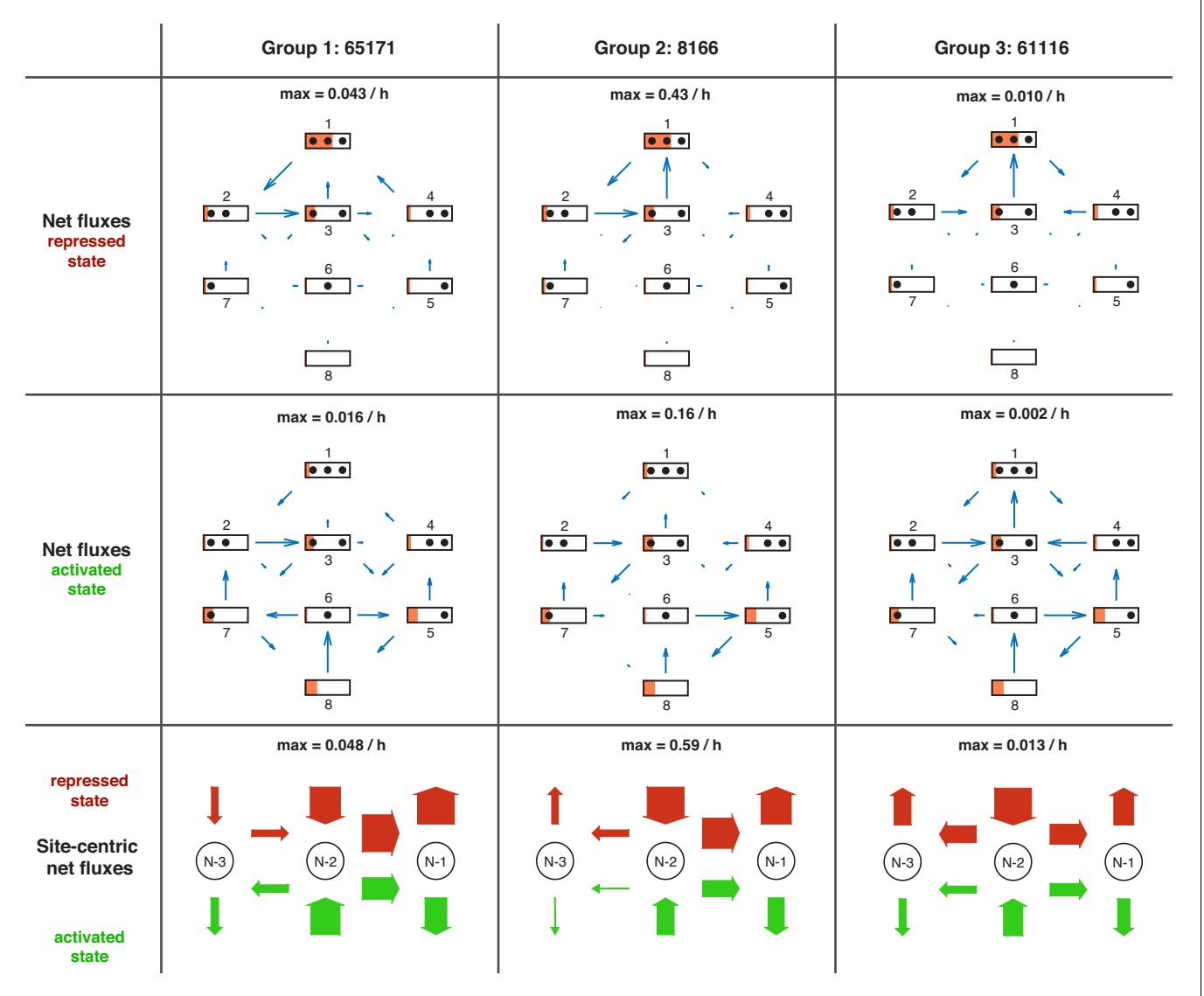

**Figure 5.** Overview of net fluxes and site-centric net fluxes for the three group representatives. First two rows: net fluxes in repressed and activated promoter state (arrows) with configuration probabilities as orange horizontal bars (a filled promoter rectangle corresponds to probability 1). Arrow length indicates the relative flux amount within a flux network with the maximum stated above. Third row: site-centric net fluxes in repressed (red) and activated (green) state, obtained by summing all assembly/disassembly net fluxes at each site and sliding net fluxes between N-1 and N-2 as well as N-2 and N-3. Here, the arrow thickness indicates the amount of flux with the maximum stated above.

The online version of this article includes the following figure supplement(s) for figure 5:

**Figure supplement 1.** Directional fluxes in repressed promoter state of all satisfactory models.

**Figure supplement 2.** Directional fluxes in activated promoter state of all satisfactory models.

**Figure supplement 3.** Net fluxes in repressed promoter state of all satisfactory models.

**Figure supplement 4.** Net fluxes in activated promoter state for each satisfactory model in stage 3.

**Figure supplement 5.** Site-centric net fluxes of all satisfactory models.

**Figure supplement 6.** Configuration distribution dynamics during chromatin opening with instantaneous signal.

**Figure supplement 7.** Configuration distribution dynamics during chromatin closing with instantaneous signal.

activated steady state for configuration three and some models also for configurations 2 and 4 (*Figure 5—figure supplement 6*). This behavior should translate into a scenario with a continuous regulation change as well. Similarly for closing, our models predict a non-monotonic probability dynamics

for configurations 3, 4, and 6 and some models also for configurations 2 and 7 (*Figure 5—figure supplement 7*).

### Extending the threshold

The properties of satisfactory models remained stable upon increasing the logarithmic likelihood ratio threshold $R_{max}$ from 6 to 7. This yielded 28 models with $R_3 < R_{max}$ (data not shown). A notable exception was the appearance of models with both, global sliding and sliding away from N-2 (S2*) and the first model with only one sliding process (S2*). Otherwise, they all showed very similar properties as the models selected with $R_{max} = 6$.

## Discussion

The following discussion is structured into five parts. We begin by reviewing our new modeling framework, which has broader applicability, beyond the analysis presented here. We then discuss our unexpected finding of nucleosome assembly playing a role in chromatin regulation, and consider the experimentally testable predictions of our satisfactory models. Lastly we propose further applications of our modeling framework, and clarify the effects of the sticky N-3 mutation on different measurements.

### Establishment of a new effective modeling approach for nucleosome dynamics

We present a new approach for modeling nucleosome dynamics at promoters: regulated on-off-slide models. They include nucleosome assembly, disassembly and sliding and enable a combined fit of data for different promoter activation states. The hierarchical approach of global, site-specific and configuration-specific processes allowed us to represent different network topologies, that is, switching 'off' reactions by using a very low rate of a non-global process, and continuous variations among network topologies (*Figure 2A–C*). We investigated all regulated on-off-slide models with up to seven fitted parameters, establishing an unbiased modeling approach that can provide insight into aspects of nucleosome dynamics that are so far not directly accessible to experiments.

Using the *PHO5* promoter as example, regulated on-off-slide models provided a unique integration of four different data sets in nucleosome resolution: nucleosome configuration measurements, our own nucleosome accessibility experiments in sticky N-3 mutants and two Flag/Myc-tagged histone H3 exchange experiments (*Figure 2D*). Only using the configuration measurements of *Brown et al., 2013* was not enough to restrict our model set. For instance, the best and second best models after stage 1 had almost identical likelihood to reproduce the data, but very different properties (see 'Promoter configuration statistics - Model heterogeneity').

Keeping these four data sets in mind, we designed our models to consider full nucleosomes, not individual histones, during assembly and disassembly and used an effective description rather than a more detailed approach with base-pair resolution and transcription factor dynamics (*Kharerin et al., 2016*). In this way, steady state data combined with dynamical data provided new insights into the nucleosome configuration dynamics, for example net fluxes between nucleosome configurations (*Figure 5*). We arranged our regulated on-off-slide models such that at least one process was regulated, that is, its rate value varied depending on the promoter state. This gave us the opportunity to examine how regulation between different promoter states was most likely achieved. We found that regulation from repressed over weakly activated to activated promoter states can be surprisingly simple, only using one regulated process, while the rates of all other processes remain constant.

Out of 68,145 tested models only seven models satisfied our threshold criterion after fitting all four data sets. All seven satisfactory models enabled sliding away from the N-2 position, but towards the N-2 only from the N-3 position. Equilibrium models and models without any sliding within the tested class could not simultaneously fit all four data sets. As we made all sliding processes optional, this showed that sliding is essential and has a net directionality. Configuration-specific processes, that is, processes governing only a single reaction, were used in all but two of the seven models, and could increase the likelihood. In these two cases, the necessary coupling between nucleosomal positions needed to find agreement with the experiments was introduced only by sliding.

By incorporating dynamical Flag-/Myc-tagged histone exchange data sets we were able to set the time scale for each model, which can not be done by using steady state data only. This is a

completely new use case for these data sets and we took care not to over-interpret them. We found that some models had a rather 'sloppy' time scale, that is, the time scale could vary strongly without notably decreasing the fit quality. One reason was that the ratios at different times of Flag over Myc amount at N-1 needed to be shifted by their mean to take into account the high fit error of the absolute values in *Dion et al., 2007*. The method of *Rufiange et al., 2007* did not have this problem, but unfortunately gave us measurements at only one time point, not restricting the slope of the dynamics of the ratio of N-1 and N-2 Flag amounts (*Figure 3—figure supplement 1A*).

Our models were selected according to four independent data sets derived from three orthogonal experimental approaches. Importantly, given the time scales, we could estimate lower bounds on how quickly the satisfactory models could switch from a closed state to an open state and vice versa. Taking into account the time scale errors, these effective chromatin opening rates were compatible with additional independent experimental values (see 'Properties of satisfactory models - Bounds for chromatin opening and closing times'). Furthermore, the ratios between effective chromatin opening versus closing rates were only compatible with the regulated assembly rather than disassembly models, even already after stage 1 just using the data from *Brown et al., 2013*. This amounts to confirmation by an additional fourth type of orthogonal data that was not used in the fitting procedure.

Before considering possible tests of predictions regarding Flag/Myc dynamics as well as chromatin opening and closing dynamics and further applications of our approach, we discuss the surprising role of nucleosome assembly in chromatin regulation within the satisfactory models.

## Unexpected role of nucleosome assembly in chromatin regulation

The identification of essential and directional sliding during *PHO5* promoter chromatin opening, the estimated time scales and the central role for the N-2 nucleosome in all seven models matched expectations based on earlier studies. However, we were surprised that only a regulated assembly process was compatible with the data, but not a regulated disassembly process, as the latter would be commonly expected. In case of an equilibrium system, the same chromatin opening could result from more disassembly or from less assembly, but, as mentioned above, our selected models are non-equilibrium models and the tested equilibrium models did not fit the data at all. Therefore, the selection of regulated assembly does not just represent the flip side of the common view but seems indeed surprising. We note that models with regulated disassembly could result in satisfactory fits to all data sets after we allowed one additional fitted parameter, but were still a very small minority among all satisfactory models, with best maximum likelihood drastically lower (less than $1/20$) than the best assembly regulated models (*Figure 3—figure supplement 3B*).

At first sight, regulated assembly counters the common view on promoter chromatin opening mechanisms as derived from pioneering studies at the yeast *PHO5* (*Korber and Barbaric, 2014*) or other, like the *HO* (*Cosma et al., 1999*) promoter, but also at mammalian promoters like the glucocorticoid-regulated MMTV promoter (*Deroo and Archer, 2001*; *Johnson et al., 2008*). According to this view, chromatin opening is triggered by binding of a (transcription) factor that locally recruits, either directly or via histone modifications like acetylation (*Hassan et al., 2001*), a chromatin remodeler, which in turn mediates nucleosome removal either by sliding and/or by disassembly (*Sudarsanam and Winston, 2000*; *Vignali et al., 2000*; *Workman, 2006*; *Becker and Hörz, 2002*; *Narlikar et al., 2002*). It is mainly based on (i) experiments showing physical interactions between transcription factors and remodelers (*Neely et al., 1999*; *Yudkovsky et al., 1999*; *Natarajan et al., 1999*) or between remodelers and modified chromatin (*Hassan et al., 2001*), (ii) on chromatin immunoprecipitation or microscopy data showing transcription factor or histone modifier recruitment to the promoter upon promoter activation (*Cosma et al., 1999*; *Kowenz-Leutz and Leutz, 1999*; *Barbaric et al., 2003*; *Dhasarathy and Kladde, 2005*; *Johnson et al., 2008*), and (iii) on in vitro assays demonstrating nucleosome sliding and disassembly activities for chromatin remodelers (*Tsukiyama et al., 1994*; *Lorch et al., 1999*; *Längst et al., 1999*; *Hamiche et al., 1999*). As chromatin opening amounts to the net removal of nucleosomes it was always intuitive to see nucleosome disassembly as the regulated process. Nonetheless, we note that remodelers were equally shown to mediate nucleosome assembly in vitro (*Lorch et al., 1999*; *Haushalter and Kadonaga, 2003*) and wonder if the well-documented remodeler recruitment at promoters upon promoter activation may also result in downregulation of nucleosome assembly rather than upregulation of disassembly.

Before ATP dependent remodeling enzymes and their active nucleosome displacement activities were fully recognized, it was proposed that binding competition between sequence specific DNA binders like transcription factors and the histone octamer, that is, mass action principles, was a major mechanism for nucleosome removal. For example, if a nucleosome was positioned over Gal4 binding sites in the absence of Gal4 it could be displaced by adding Gal4 in vitro (*Workman and Kingston, 1992*) or inducing Gal4 in vivo (*Morse, 1993*). Recently, the class of pioneer factors and general regulatory factors that have important roles in opening chromatin or keeping chromatin open, for example in the context of cellular reprogramming, were also suggested to displace nucleosomes by binding competition (*Yan et al., 2018*; *Donovan et al., 2019*; *Iwafuchi et al., 2020*). For the *PHO5* promoter, this mechanism seemed attractive as its transactivator Pho4 indeed competes with nucleosome N-2 during binding to the intranucleosomal UASp2. Further, a *PHO5* promoter lacking UASp2, that is, left only with the constitutively accessible UASp1 in-between N-2 and N-3, was not induced during phosphate starvation in vivo (*Ertel et al., 2010*).

However, early studies dismissed an essential role for binding competition at the *PHO5* promoter as (i) the coregulated *PHO8* promoter was opened without an intranucleosomal UASp (*Barbarić et al., 1992*), (ii) even the ΔUASp2 *PHO5* promoter mutant could be opened if *PHO4* was overexpressed (*Ertel et al., 2010*) and (iii) even an overexpressed Pho4 version containing a functional DNA-binding but no transactivation domain could not displace N-2 and trigger chromatin opening (*Svaren et al., 1994*) although just the Gal4 DNA binding domain could displace nucleosomes in other contexts (*Workman and Kingston, 1992*; *Morse, 1993*). However again, binding competition at UASp2 in N-2, while not essential if binding to UASp1 was boosted by increasing its affinity or by *PHO4* overexpression, did have a supportive role in *PHO5* promoter chromatin remodeling for wild type UASp1 and *PHO4* expression levels (*Ertel et al., 2010*).

Therefore, we find it plausible that the downregulation of nucleosome assembly implicated by our models could result from the competition between Pho4 binding to the promoter and nucleosome assembly at the promoter. This pertains only to the wild type version of the *PHO5* promoter, with Pho4 binding to UASp elements, which we exclusively studied here. Impeding the assembly of N-2 by Pho4 binding at UASp2 could directly correspond to the downregulated N-2 assembly processes in our models 27,443 and 27,448 (*Figure 4*). Pho4 binding to both UASp2 and UASp1 may correspond to impaired global assembly in the other models.

The emphasis on regulation of nucleosome assembly rather than disassembly within our models thus suggests a new perspective on the effective dynamics of chromatin remodeling during promoter opening and calls for experimental assessment in future studies. These may address the recently revived debate about the role of binding competition in nucleosome remodeling.

## Experimentally testable predictions

To further assess the validity of our satisfactory models, we suggest several lines of studies to investigate predictions regarding the nucleosome dynamics in the steady states as well as during chromatin opening and closing. Firstly, to obtain more information about the dynamics within the steady states, more Flag-/Myc-tagged histone exchange experiments would be most helpful. Measured Flag amount ratios between N-3 and N-2 or N-3 and N-1 and more time points for each ratio would allow further discrimination between our models. Six of our seven satisfactory models predict a similar Flag amount between N-3 and N-1 (*Figure 3—figure supplement 1B*), while the satisfactory models exhibit very different behavior for the ratio between N-3 and N-2 (*Figure 3—figure supplement 1C*). Additionally these measurements could in principle also be taken in the activated promoter state, where our remaining models also show diverse behavior (*Figure 3—figure supplement 1D,E,F*).

A second line of studies may investigate the opening and closing dynamics in the wild-type scenario in more detail. We argued that the experimentally observed faster closing than opening is consistent with regulation by assembly and inconsistent with regulation by disassembly (see 'Properties of satisfactory models - Bounds for chromatin opening and closing times'). The underlying reason for the faster closing, however, remains unclear. It might be due to faster nucleosome closing dynamics, as our models suggest, but possibly also due to faster signaling speed in closing and a delayed signaling in opening. To address this question and also obtain further detailed information we suggest a time series of configuration measurements for chromatin opening and closing. The measured distributions of configurations at different times can then be compared with our models' predictions,

especially the non-monotonic dynamics of certain configurations (*Figure 5—figure supplement 6* and *Figure 5—figure supplement 7*). Prerequisite for these studies will be the generation of single molecule nucleosome configuration data as in *Brown et al., 2013* at different time points of chromatin opening. It should be noted that such data are nowadays much easier to obtain than by the psoralen-TEM approach of *Brown et al., 2013* due to the recent development of single molecule long-read sequencing of DNA methylation footprints (*Shipony et al., 2020*; *Stergachis et al., 2020*; *Lee et al., 2020*; *Oberbeckmann et al., 2019*).

## Future applications of our newly established approach

In the following, we propose further applications of our modeling approach in studies with manipulations which possibly lead to dynamics diverging from the wild-type behavior. Indeed, a major outcome of our study is that our approach allows to derive which kind of effective dynamics underlie different chromatin states and the transition between states. This was so far hardly accessible as mostly only the occupancy at sites at certain conditions was determined and compared. It may well be that seemingly the same 'open *PHO5* promoter' as judged by chromatin accessibility in nuclease-based or by chromatin bound factors in chromatin immunoprecipitation assays does correspond to different configuration dynamics or was generated along different trajectories. While former studies concluded that the *PHO5* promoter 'could still be opened' despite certain manipulations, for example, lack of replication, lack of USAp2, lack of Snf2, Gcn5 or other chromatin cofactors, our approach may reveal that the remodeling pathways and initial or final chromatin states were affected and quite different nonetheless.

Manipulations that are likely to change the dynamics and the underlying biological processes give us no information on the wild type dynamics unless we already know the effect of the manipulation and are able to model it within our modeling approach (as we assume for instance for the sticky N-3 mutants in 'Integration of *PHO5* promoter mutant data'). If the effect of the manipulation cannot be incorporated within our models, the resulting datasets cannot be used for a combined fit with the existing wild-type data. Nevertheless, manipulated promoter datasets can be fitted from scratch with our regulated on-off-slide models, if the needed data is obtained (configurations under repressed and activating conditions, Flag-/Myc-tagged histone exchange measurements and possibly also effects of further sticky mutations on configurations or occupancies). We then obtain a picture of the effective dynamics in the manipulated strains which then can be compared to the wild type. In this way, manipulations that earlier were found to also allow chromatin opening and thus were deemed unimportant, might still reveal different steady state configuration distributions and effective nucleosome dynamics.

A first manipulation study might employ Pho4 binding site mutations with possibly altered positions to affect the hypothesized binding competition, for example a ΔUASp2 *PHO5* promoter mutant without intranucleosomal Pho4 site under *PHO4* overexpression. In hindsight, even though *PHO5* promoter opening by *PHO4* overexpression in the absence of UASp2 led to the same open promoter state as judged by DNAseI indirect endlabeling and restriction enzyme accessibilities (*Ertel et al., 2010*), different effective nucleosome dynamics may underlie this mutant compared to the wild type promoter. If single molecule nucleosome configurations were available for this ΔUASp2 *PHO5* promoter chromatin transition, our modeling approach could test this. Another approach is moving the intranucleosomal Pho4 site to the N-1 or N-3 nucleosome, either instead or in combination with the native site in the N-2 nucleosome to investigate the effects of the positions of the intranucleosomal Pho4 site on the regulated processes of the fitted models.

A second manipulation study might investigate remodeler deletion strains to shed light on the influence of remodeler actions on the effective nucleosome dynamics. Nucleosome movement in vivo is mediated by ATP dependent remodelers and histone modifying enzymes. Five different remodelers (SWI/SNF, RSC, INO80, Isw1, Chd1) (*Barbaric et al., 2007*; *Musladin et al., 2014*) and at least three histone modifiers (Gcn5, Rtt109, Set1) (*Barbaric et al., 2001*; *Korber et al., 2006*; *Carvin and Kladde, 2004*) are involved in opening *PHO5* promoter chromatin. These cofactors are redundant, that is, single-deletion mutants are still inducible (*Barbaric et al., 2007*). However, comparing the effective nucleosome dynamics at the *PHO5* promoter as determined by our approach in mutant backgrounds where one or several of these factors are missing should reveal different fluxes and net fluxes. Furthermore one could test if a lack of Gcn5, the main histone acetyltransferase that generates the acetylated lysine residues involved in recruiting SWI/SNF and RSC via their

bromodomains, has a similar effect as removing these remodeling enzymes (*Hassan et al., 2002*; *VanDemark et al., 2007*).

A third manipulation study could explore the effective dynamics without the influence of replication and investigate the role of replication during *PHO5* promoter opening. Induction via phosphate starvation or via the *pho80* allele occurs during ongoing cell cycle with intermittent S phases where nucleosomes are globally disassembled and reassembled. Induction via phosphate starvation or via the *pho80* allele occurs during the ongoing cell cycle with intermittent S phases which include global nucleosome disassembly and reassembly. It was shown that *PHO5* promoter chromatin can be opened without replication in arrested cells (*Schmid et al., 1992*), but our approach can now assess if this changes the effective nucleosome configuration dynamics.

### Clarification of effects of the sticky N-3 mutation

While we obtained and used the same two sticky N-3 mutant strains as published (*Small et al., 2014*), we did not use their published chromatin data. These were generated by an at that time novel single molecule DNA methylase footprinting approach and seemed less trustworthy for the following reasons. First, it was not clear if the DNA methylation reactions were saturated (for a detailed discussion see *Oberbeckmann et al., 2019*). Second, accessibility at the N-3 position in wild-type cells was lower (7%) in the activated compared to the repressed (28%) state, which is in contrast to chromatin opening upon activation and the data by *Brown et al., 2013* with 55% accessibility at N-3 in the activated promoter state. Third, the authors did not detect any case where all three N-1 to N-3 nucleosomes were removed upon activation, although this configuration was among the most populated in the study by *Brown et al., 2013*. Both the second and third point may be due to incomplete methylation and/or to erroneous interpretation of methylation footprint patterns, as not only nucleosomes but also other factors like the transcription initiation machinery could obstruct methylation. Fourth, we found it hard to believe that the point mutations introduced in the sticky N-3 *PHO5* promoter mutant versions would completely abolish chromatin opening not only at the N-3 but also at the N-2 nucleosome as claimed by the authors (*Small et al., 2014*).

Therefore, we rather employed the classical and well-documented restriction enzyme accessibility assay (*Almer et al., 1986*; *Gregory et al., 1999*) to monitor *PHO5* chromatin opening in the sticky N-3 mutants. In this assay, both sticky N-3 mutants still showed considerable chromatin opening but we confirmed the general conclusion by *Small et al., 2014* that opening of both the N-2 and N-3 nucleosomes was less extensive than for the wild type promoter. Even though the effect was less pronounced than claimed originally, it is remarkable how it substantiates earlier observations at the yeast *PHO8* and *PHO84* promoters (*Wippo et al., 2009*), which are co-regulated with the *PHO5* promoter, about the role of underlying DNA sequence in stabilizing nucleosomes against remodeling. Recent in vitro experiments started to reveal how chromatin remodeling enzymes are regulated by nucleosomal DNA sequences (*Lorch et al., 2014*; *Winger and Bowman, 2017*; *Krietenstein et al., 2016*). The *PHO5* promoter periodicity mutants pioneered by *Small et al., 2014* may provide an impressive case how this is relevant in vivo.

## Materials and methods

The maximum likelihood fit procedure (*Figure 2D*) was implemented in MATLAB and the source code is available at https://github.com/gerland-group/PHO5_on-off-slide_models.

### Maximum likelihood fits

Depending on the stage of the analysis, we optimized the parameter values, that is, the rate values of the processes within a given model, $\vec{r}$, by maximizing the sum of log10 likelihood values: $L_I(\vec{r})$ in stage 1, $L_I(\vec{r}) + L_{II}(\vec{r})$ in stage 2 and $L_I(\vec{r}) + L_{II}(\vec{r}) + L_{III}(\vec{r})$ in stage 3. Note that the optimal $\vec{r}$ can differ between different stages. We ignored additive constants to the log likelihood, so that after including the next data set into the fit, a perfect agreement between model and the additional data set, already with the values $\vec{r}$ from the previous stage, would lead to the same log likelihood value as before. In all stages we used the MATLAB fmincon function to find the parameter values $\vec{r}$ that maximize the likelihood.

With the rate parameter notation introduced in *Figure 2*, we have for the example of *Figure 3A*:

$$\vec{r} = (r_A^{\mathrm{rep}}, r_A^{\mathrm{w.act}}, r_A^{\mathrm{act}}, r_D, r_{D1-4}, r_{S2^*}, r_{S3-4})^\top. \tag{1}$$

Let $Q(\vec{r}^\sigma)$ be the transition rate matrix of the Markov process defined by the model for promoter state $\sigma$, where $\vec{r}^\sigma$ denotes the vector containing the regulated parameter value(s) of promoter state $\sigma$ and all constitutive parameter values. A non-diagonal entry $Q_{ij}(\vec{r}^\sigma)$ is the rate to go from configuration $i$ to configuration $j$ and is non-zero only for valid assembly, disassembly and sliding reactions and then given by the entry of $\vec{r}^\sigma$ which holds the parameter value of the process that governs this reaction in the given model. If sliding reactions are not governed by any sliding process within the model, their rate is set to zero. Diagonal entries are given by $Q_{ii}(\vec{r}^\sigma) = -\sum_{j \neq i} Q_{ij}(\vec{r}^\sigma)$. In the example of **Figure 3A**, the transition rate matrix in the activated state is given by (with '...' representing the diagonal entries)

$$Q(\vec{r}^{\mathrm{act}}) = \begin{pmatrix} \dots & r_D & r_D & r_{D1-4} & 0 & 0 & 0 & 0 \\ r_A^{\mathrm{act}} & \dots & r_{S2^*} & 0 & 0 & r_D & r_D & 0 \\ r_A^{\mathrm{act}} & 0 & \dots & r_{S3-4} & r_D & 0 & r_D & 0 \\ r_A^{\mathrm{act}} & 0 & r_{S2^*} & \dots & r_D & r_D & 0 & 0 \\ 0 & 0 & r_A^{\mathrm{act}} & r_A^{\mathrm{act}} & \dots & 0 & 0 & r_D \\ 0 & r_A^{\mathrm{act}} & 0 & r_A^{\mathrm{act}} & r_{S2^*} & \dots & r_{S2^*} & r_D \\ 0 & r_A^{\mathrm{act}} & r_A^{\mathrm{act}} & 0 & 0 & 0 & \dots & r_D \\ 0 & 0 & 0 & 0 & r_A^{\mathrm{act}} & r_A^{\mathrm{act}} & r_A^{\mathrm{act}} & \dots \end{pmatrix}. \tag{2}$$

The steady state distribution $p_{i\sigma}$ of $Q(\vec{r}^\sigma)$ is the solution of $p_{j\sigma} = \sum_i p_{i\sigma} Q_{ij}(\vec{r}^\sigma) = 0$.

Then $L_I(\vec{r})$ can be calculated using the multinomial distribution,

$$L_I(\vec{r}) = \sum_\sigma \log_{10} \left[ \frac{(\sum_i n_{i\sigma})!}{\prod_i n_{i\sigma}!} \prod_i p_{i\sigma}^{n_{i\sigma}} \right], \tag{3}$$

with $n_{i\sigma}$ being the number of observations of the corresponding promoter configurations (**Figure 1—source data 1**).

For each model, we used 100 different sets of initial parameter values to ensure a robust maximum. To calculate the steady state distribution of a given model for fixed parameter values, we used the state reduction algorithm (**Sheskin, 1985**; **Grassmann et al., 1985**; **Shanbhag and Rao, 2003**). Alternatively, the Matrix-Tree theorem can be used to find the steady state distributions of Markov processes (**Wong and Gunawardena, 2020**). We limited the range of parameter values to $[10^{-2}; 10^2]$, with one being the rate value of the global assembly process for the activated state. A wider range of $[10^{-3}; 10^3]$ did not affect the results. In 3.6% of all models the 100 tries found at least two different maximal likelihood values, which were always extremely low. In 2.4% of all models, the found maximum likelihood parameter values were not unique. In both cases, none of these problematic models were among models with relatively high maximal likelihoods.

In stage 2, the sum of $L_I(\vec{r})$ and $L_{II}(\vec{r})$ is optimized. Assuming the experimental fold changes are normally distributed the log10 likelihood of a model to reproduce the new data up to an additive constant is given by

$$L_{II}(\vec{r}) \ln(10) = \sum_{m=1}^{2} \max_{\vec{\kappa}^m} \sum_{s=2}^{3} - \frac{\left( f_s(\vec{r}, \vec{\kappa}^m) - f_{sm}^{\mathrm{mean}} \right)^2}{2 f_{sm}^{\mathrm{var}}}, \tag{4}$$

with $f_{sm}^{\mathrm{mean}}$ and $f_{sm}^{\mathrm{var}}$ being mean and variance, respectively, of the measured accessibility fold changes in active state of sticky N-3 mutant $m$ at nucleosome site $s$ (two for N-2 and 3 for N-3), $f_s(\vec{r}, \vec{\kappa}^m)$ being the corresponding model fold change, and $\vec{\kappa}^m$ being the values of the rate prefactors of sticky mutant $m$.

To obtain $f_s(\vec{r}, \vec{\kappa}^m)$ for each model, we calculated a modified transition rate matrix for each mutant using the non-diagonal part of the transition rate matrix $Q(\vec{r}^{\mathrm{act}})$ and multiplied it component-wise with the matrix $W(\vec{\kappa}^m)$ containing the prefactor values $\vec{\kappa}^m$ for the affected reactions for mutant $m$ (and one otherwise). Using the modified transition rate matrices, we calculated the mutant steady state distributions and finally the corresponding fold ratios of accessibilities at N-2 and N-3.

We used four prefactors per sticky N-3 mutant, one for each group of reactions, assembly at N-3, disassembly at N-3, sliding from N-3 to N-2 and from N-2 to N-3 (*Figure 2—figure supplement 5*), respectively, leading to $\vec{\kappa}^m = (\kappa^m_{a3}, \kappa^m_{d3}, \kappa^m_{s23}, \kappa^m_{s32})^\top$. The off-diagonal part of $W(\vec{\kappa}^m)$ is then given by

$$
W(\vec{\kappa}^m) = \begin{pmatrix}
\ldots & 1 & 1 & \kappa^m_{d3} & 1 & 1 & 1 & 1 \\
1 & \ldots & 1 & 1 & 1 & \kappa^m_{d3} & 1 & 1 \\
1 & 1 & \ldots & \kappa^m_{s32} & \kappa^m_{d3} & 1 & 1 & 1 \\
\kappa^m_{a3} & 1 & \kappa^m_{s23} & \ldots & 1 & 1 & 1 & 1 \\
1 & 1 & \kappa^m_{a3} & 1 & \ldots & 1 & 1 & 1 \\
1 & \kappa^m_{a3} & 1 & 1 & 1 & \ldots & \kappa^m_{s23} & 1 \\
1 & 1 & 1 & 1 & 1 & \kappa^m_{s32} & \ldots & \kappa^m_{d3} \\
1 & 1 & 1 & 1 & 1 & 1 & \kappa^m_{a3} & \ldots
\end{pmatrix},
\tag{5}
$$

where the diagonal does not matter due the component-wise multiplication with the non-diagonal part of $Q(\vec{r}^{\mathrm{act}})$ in order to obtain the non-diagonal part of the mutant rate transition matrices. Thus, $L_{II}$ depends only on $\vec{r}^{\mathrm{act}}$. Note that the exact values of prefactors found during optimization depended on their initial condition, as their best values were often sloppy or not unique, but still resulted in the maximum likelihood.

For stage 3, $L_{III}$ has two contributions, one for each histone H3 exchange experiment:

$$
L_{III}(\vec{r}) = L^1_{III}(\vec{r}) + L^2_{III}(\vec{r})
\tag{6}
$$

Strictly speaking $L_{III}$ depends only on $\vec{r}^{\mathrm{rep}}$, since all the histone H3 exchange experiments were done in the repressed state. For the first contribution, to fit the data from *Rufiange et al., 2007*, we used

$$
L^1_{III}(\vec{r})\ln(10) = -\frac{(g(\vec{r},t') - g^{\mathrm{mean}})^2}{2g^{\mathrm{var}}}
\tag{7}
$$

with $g^{\mathrm{mean}}$ and $g^{\mathrm{var}}$ being the mean and variance, respectively, of the measured log2 ratios of Flag amounts at N-1 over N-2 (Flag-H3 MNase-ChIP in *Rufiange et al., 2007*, ratio values 0.591 and 0.483 for replicate 1 and 2, respectively) and $g(\vec{r},t')$ the corresponding log2 ratio of the model (see Materials and methods section: H3 histone exchange model) for measurement time $t' = 2\,\mathrm{h}$ (not corrected for the lag time).

For the second contribution, let $h^{\mathrm{mean}}_j$ denote the measured normalized mean log2 ratios of Flag amount over Myc amount at N-1, with $j = 1, 2, 3, 4$ indicating the four different time points. We obtained $\vec{h}^{\mathrm{mean}} = (-0.417, 1.24, 1.87, 2.60)^\top$ from *Dion et al., 2007* as follows: we recalculated the normalization constant of each time point using the measured mean log2 Myc/Flag signal ratios as described (supplementary material of *Dion et al., 2007* using the whole-genome commercial microarrays (Agilent) data with the nucleosome pool parameters as in the section above) and then took the normalized results of the probe at the N-1 position of the *PHO5* promoter (chr2:431049–431108). Unfortunately, neighboring probes were only in linker regions between promoter nucleosome positions. As mentioned in *Dion et al., 2007* and corroborated by our own calculations, the values $h^{\mathrm{mean}}_j$ have large uncertainties, mostly due to an additive sloppy global normalization constant leading to systematic errors, while the differences between time points were determined with reasonable accuracy. Thus, we decided to fit the measured values only after a transformation that eliminates the sloppy global normalization constant by choosing the average over the four time points as a reference: $\tilde{h}^{\mathrm{mean}}_j = h^{\mathrm{mean}}_j - 1/4\sum^4_{k=1} h^{\mathrm{mean}}_k$. Let $C$ be the resulting covariance matrix after this linear transformation, assuming an independent estimated experimental standard deviation of 0.4 before the transformation. This estimate was informed by the standard deviation of the *Rufiange et al., 2007* data as well as from perturbations of the nucleosome pool parameters when recalculating the normalization constants. The corresponding normalized values of the model are denoted by $\tilde{h}_j(\vec{r})$, calculated from the log2 ratios of Flag amount over Myc amount at N-1, $h(\vec{r}, t_j)$ (see Materials and methods section: H3 histone exchange model), using the same transformation, with $t_j$ denoting the measurement time points. Since the four measurements at different time points were linearly mapped to always have an average of zero, the estimated density follows a degenerate

multivariate normal distribution with the covariance matrix $C$. Thus the log10 likelihood can be calculated with

$$L_{III}^2(\vec{r})\ln(10) = -\frac{1}{2}\sum_{j=1}^{4}\sum_{k=1}^{4}(\tilde{h}_j(\vec{r})-\tilde{h}_j^{\text{mean}})C_{jk}^+(\tilde{h}_k(\vec{r})-\tilde{h}_k^{\text{mean}}),\tag{8}$$

where $C^+$ is the Moore-Penrose inverse (pseudoinverse) of $C$.

## H3 histone exchange model

To obtain the Flag and Myc amounts in a given model with given parameter values and then determine $g(\vec{r},t')$ and $h(\vec{r},t_j)$, we used the histone pool and nucleosome turnover models in *Dion et al., 2007* and assumed that the Myc H3 and Flag H3 amounts in the histone pool are given by $M(t)=\frac{\alpha_M}{\beta_M}$ and

$$F(t)=\begin{cases}0,\text{for }t<t_0\\\frac{\alpha_F}{\beta_F}(1-e^{\beta_F(t-t_0)}),\text{ for }t\geq t_0,\end{cases}\tag{9}$$

where we used the production rates $\alpha_F=50\,/\min$, $\alpha_M=10\,/\min$, the degradation rates $\beta_F=0.01\,/\min$, $\beta_M=0.03\,/\min$ and the lag time $t_0=15\min$ which were fitted in *Dion et al., 2007*. For $t>t_0$, the probability that a newly assembled nucleosome contains a Flag H3 is given by

$$\begin{aligned}P^+(t|N) &=\frac{F(t)}{F(t)+M(t)}\\&=1/\left(1+\frac{\alpha_M\beta_F}{\alpha_F\beta_M}/\left(1-e^{-\beta_F(t-t_0)}\right)\right)\\&\to 1/\left(1+\frac{\alpha_M\beta_F}{\alpha_F\beta_M}\right)=0.9375,\text{for }t\to\infty.\end{aligned}\tag{10}$$

In *Dion et al., 2007*, the conditional probability that a given nucleosome at site $l$ at time $t$ contains a Flag H3 then fulfills the ordinary differential equation

$$\frac{d}{dt}P_l(t|N)=\lambda_l(P^+(t|N)-P_l(t|N)),\tag{11}$$

with $\lambda_l$ being an effective turnover rate at probe position $l$. In our case, the dynamics of the three promoter nucleosomes are coupled, determined by the transition rate matrix $Q(\vec{r}^\sigma)$ of a given regulated on-off-slide model. At this stage, we included different nucleosome types (i.e. Flag and Myc) into the model, replacing the eight promoter configurations by all 27 possibilities to arrange no, a Flag or a Myc nucleosome at each of the three sites. Based on $Q(\vec{r}^\sigma)$ and $P^+(t|N)$, we define an extended Flag/Myc transition rate matrix $E(\vec{r}^\sigma,P^+(t|N))$. Each 'new' assembly reaction rate in $E(\vec{r}^\sigma,P^+(t|N))$ is given by the corresponding 'old' assembly rate in $Q(\vec{r}^\sigma)$ times either $P^+(t|N)$ or $1-P^+(t|N)$, for a new Flag or Myc nucleosome, respectively. To find the corresponding 'old' reaction any extended Flag/Myc configuration is projected to one of the eight normal nucleosome configurations simply by ignoring the Flag/Myc tag information. For example, denoting Flag- and Myc-tagged nucleosomes with 'F' and 'M', respectively, an assembly reaction from the state (F, M, 0) to the state (F, M, M) in the extended model corresponds to an assembly reaction from state (1, 1, 0) to the state (1, 1, 1) in the normal model, and its reaction rate is multiplied by $1-P^+(t|N)$ in the extended model, since the new nucleosome is Myc-tagged. The rates of sliding and disassembly of Flag or Myc nucleosomes are assumed to be equal to the corresponding normal sliding and disassembly rates. The probability of extended configuration $i$ at time $t$ is the $i$-th entry of $\vec{q}^*(t)$, the solution of

$$\frac{\partial}{\partial t}\vec{q}=E^\top(\vec{r}^\sigma,P^+(t|N))\vec{q},\tag{12}$$

where $\sigma$ is fixed in the repressed state, in which all histone exchange experiments took place. The log2 ratios of Flag at N-1 over Flag at N-2 amount, $g(\vec{r},t)$, and Flag over Myc amounts at N-1, $h(\vec{r},t)$, of each model then correspond to log2 ratios of sums of $q_i^*(t)$ over suitable configurations $i$ with Flag or Myc nucleosomes at the wanted sites.

## Sensitivity analysis

In order to determine how sloppy the found best parameter values for a given model are, we performed a simple sensitivity analysis, by calculating the log10 likelihood $L_I + L_{II} + L_{III}$ along certain directions from the best fit point in logarithmic parameter space. We found that an approximation of the real likelihood function by a second-order Taylor expansion at the best fit point worked only in a small area, as expected in a highly non-linear setting, but too small to determine parameter sloppiness properly.

As a compromise between properly scanning the parameter space and computational feasibility, we chose a small number of test directions: each fitted parameter value individually, the eigenvectors of the numerical Hessian of the likelihood function at the best fit, as well as the numerical gradient, which can be non-zero if the best fit point lies on the boundary. We ignored the boundary during the sensitivity analysis, to also take into account sloppiness that 'reaches over' the boundary. Along these directions, we tested in exponentially increasing steps from the best fit position which positions in parameter space lead to a decrease of the likelihood by $\approx 50\%$, that is, a log10 likelihood ratio change of $\approx 0.30$, which is of similar order as the log10 likelihood differences within our group of satisfactory models. We then obtained 'error bars' for each parameter by taking the largest deviation of the log10 parameter value at the 50% likelihood level from the best value found in all tested directions (*Figure 4—source data 1* and *Figure 4—source data 2*).

## Effective chromatin opening and closing rates

The effective trajectory in time of the regulated process rate from the value of the repressed state to the value of the activated state depends on how fast the cell senses the phosphate starvation and subsequent signal processes. To obtain a reasonable upper bound for the chromatin opening rate, we assumed the regulation happens instantaneously, that is, the activated rate value of the regulated processes applies immediately at the change of the medium for a population in repressed state. Then the promoter configuration distribution decays exponentially toward the activated steady state with a rate well approximated by the negative eigenvalue of the transition rate matrix closest (but not equal) to zero, taking into account the fitted time scale. This 'effective chromatin opening rate' is an upper bound of how fast a given model can switch to the activated state. Conversely, we did the same calculations for the 'effective chromatin closing rate', which is an upper bound of how fast a given model can switch to the repressed state.

## Sticky N-3 experiments

Strains 'sticky N-3 mutant 1' and 'sticky N-3 mutant 2' used for restriction enzyme accessibility assays were generated by transformation of linear fragments of plasmids ECS53 and ECS56, respectively, into the wild type strain BY4741 as described for the 'periodicity mutants' in *Small et al., 2014*. For the sticky N-3 mutant 1, the sequence GTTTTCTCATGTAAGCGGACGTCGTC inside the *PHO5* promoter was replaced with GTTTTCTTATGTAAGCTTACGTCGTC. For sticky N-3 mutant 2, GCGCAAA TATGTCAACGTATTTGGAAG was replaced with GCGCAAATATGTCAAAGTATTTGGAAG. Strains were grown in YPDA medium to logarithmic phase for repressive (+Pi) and shifted from logarithmic phase to phosphate-free YNB medium (Formedia) over night for inducing (-Pi) conditions. Nuclei preparation, restriction enzyme digestion, DNA purification, secondary digest, agarose gel electrophoresis, Southern blotting, hybridization, and Phosphorimager analysis were as in *Musladin et al., 2014*. Secondary digest was with HaeIII for both ClaI and HhaI digests probing N-2 or N-3, respectively. The probe for both ClaI and HhaI digests corresponded to the ApaI-BamHI restriction fragment upstream of N-3.

## Acknowledgements

We thank Eliza Small (Northwestern University Feinberg School of Medicine, Chicago, USA) for providing the plasmids ECS53 and ECS56. Furthermore we thank Oliver J Rando and his group for providing the fitted nucleosome pool parameters used in *Dion et al., 2007* and Amine Nourani and his group for providing the Flag-H3 MNase-ChIP values for the N-1 and N-2 sites of the two replicates in *Rufiange et al., 2007*. MRW is a member of the Graduate School of Quantitative Biosciences Munich (QBM).

## Additional information

### Funding

| Funder | Grant reference number | Author |
|---|---|---|
| Deutsche Forschungsgemeinschaft | SFB863 | Ulrich Gerland |
| Deutsche Forschungsgemeinschaft | SFB1064 | Philipp Korber |

The funders had no role in study design, data collection and interpretation, or the decision to submit the work for publication.

### Author contributions

Michael Roland Wolff, Conceptualization, Software, Formal analysis, Validation, Investigation, Visualization, Methodology, Writing - original draft, Writing - review and editing; Andrea Schmid, Validation, Investigation; Philipp Korber, Conceptualization, Resources, Supervision, Funding acquisition, Validation, Investigation, Methodology, Writing - review and editing; Ulrich Gerland, Conceptualization, Resources, Supervision, Funding acquisition, Methodology, Project administration, Writing - review and editing

### Author ORCIDs

Michael Roland Wolff  https://orcid.org/0000-0002-4677-4023
Philipp Korber  https://orcid.org/0000-0001-7526-6549
Ulrich Gerland  https://orcid.org/0000-0002-0859-6422

### Decision letter and Author response

Decision letter https://doi.org/10.7554/eLife.58394.sa1
Author response https://doi.org/10.7554/eLife.58394.sa2

## Additional files

### Supplementary files

• Transparent reporting form

### Data availability

All data generated or analysed during this study are included in the manuscript and supporting files. Source data files have been provided for Figure 1, Table 1 and Figure 4. Program code has been uploaded to github (see Materials and methods).

The following previously published datasets were used:

| Author(s) | Year | Dataset title | Dataset URL | Database and Identifier |
|---|---|---|---|---|
| Dion MF, Kaplan T, Kim M, Buratowski S, Friedman N, Rando OJ | 2007 | Dynamics of Replication-Independent Histone Turnover in Budding Yeast | https://www.ncbi.nlm.nih.gov/geo/query/acc.cgi?acc=GSE6666 | NCBI Gene Expression Omnibus, GSE6666 |
| Rufiange A, Jacques PÉ, Bhat W, Robert F, Nourani A | 2007 | Genome-Wide Replication-Independent Histone H3 Exchange Occurs Predominantly at Promoters and Implicates H3 K56 Acetylation and Asf1 | https://www.ncbi.nlm.nih.gov/geo/query/acc.cgi?acc=GSE8299 | NCBI Gene Expression Omnibus, GSE8299 |

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
