## [Decision Letter]

Thank you for submitting your work entitled "Effective dynamics of nucleosome configurations at the yeast *PHO5* promoter" for consideration by *eLife*. Your article has been reviewed by three peer reviewers and the evaluation has been overseen by Naama Barkai as the Senior and Reviewing Editor.

There are some issues that need to be addressed before acceptance, as outlined below:

As you will see below, the reviewers appreciated the work and the simplified theoretical treatment. However, the reviewers have concerns about the gap between the modelling results and the biological discussion. It is important that the Discussion will be revised to be more a simplified and "user friendly" Discussion and highlight key experimental predictions. in particular, it is important to explain in more detail how the "assembly-centric" suggestion in the Discussion may be experimentally tested in the light of your models.

In addition, please revise the paper to account for all points mentioned in the point-to-point reviews below.

Reviewer #1:

The manuscript under consideration puts forward a beautiful Markovian model of promoter regulation via nucleosome assembly/disassembly/sliding. The model is applied to the yeast *PHO5* promoter, for which experimental measurements are available and are being used to specify the final models from the space of possible ones. The final models agree on certain features, and these constitute the result of the analysis. The modeling process is clean and parsimonious. The authors' claim of an unbiased process is indeed convincing in that only likelihood distinguishes from among many alternatives.

There is one issue which requires clarification. The likelihood ratio is used to select preferable models. However, the models compared have different number of parameters. Why not use a penalized likelihood, e.g., BIC or AIC? The underlying problem is that the large number of models has a downside: While it avoids modeling bias, it introduces an unwanted multiple testing problem.

Overall, the paper is technically very convincing and the integration of the experimental data into the modeling process is excellent.

Figures are clear and informative.

In terms of language, the authors speak of "fit parameter". It would be better to use "fitted" to avoid possible confusion with "fit" like in "fitness".

Reviewer #2:

In this work, authors study dynamics nucleosome configurations in the promoter region of yeast *PHO5* gene. Authors examine published experimental data describing positioning of three nucleosomes in *PHO5* promoters. A few years ago, experiments by Brown et al. and Small et al. have quantified probabilities of finding eight possible nucleosome configurations when the gene is in different states like active or repressed. There are a few existing mathematical and computational models that examine *PHO5* gene regulation in the context of this data.

In this work, authors present a comprehensive and systematic approach towards enumerating all possible models, given the data, and have classified the models based on their agreement with the experimental data. Authors consider dynamics of nucleosomes among the 8 discrete states via assembly, disassembly and sliding of the 3 nucleosomes in the promoter region. They show that there are seven models (out of several thousand "models") that agree well with the data. They discuss the interesting features observed in these 7 models.

Even though earlier studies have attempted a similar approach, this is a nice work as it is comprehensive and systematic. However, there are some concerns and some suggestions that could improve the paper.

(1) Authors account only those eight promoter nucleosome configurations that are presented in the Brown et al/Small et al. experiments. However, to understand gene regulation, binding of other transcription factors (like *Pho4*) too are crucial. Binding of these factors are coupled with nucleosome dynamics. Can the model provide biologically relevant predictions without accounting for these factors? This is not clear from the current manuscript.

(1a) Authors find that certain sliding events are crucial. However, binding of *Pho4* would provide steric hinderance and prevent many of the sliding moves. How would you reconcile this given the results presented here.

(1b) Are there known mechanisms for directed sliding of nucleosomes? It is mentioned that the sliding speed is consistent with SWI/SNF, RSC remodelling complexes. But are any of these enzymes known to slide nucleosomes in a specific direction?

(2) Even though experiments have measured positioning of only three nucleosomes, isn't it possible that the positioning of these nucleosomes is influenced by the 4th nucleosome (N-4) or even the +1 nucleosome? It is known that positioning of neighbouring nucleosomes are correlated (due to their steric exclusion). Will extending the model with the 4th nucleosome (or +1 nucleosome) provide more insights about the dynamics of the 3 observed nucleosomes? If possible, that would be a new contribution beyond all the existing models.

(3) Given that many studies on this topic exist, what are the new testable predictions from this study? Please write a small subsection on the new testable predictions.

(4) Some suggestions to bring more clarity to the manuscript:

(4a) It is good to clearly define what is a "model". In this context, typically, a master equation would define a model. When authors talk about several models, it is good to mention how precisely they define a model. Readers need to understand how did authors enumerate all the models and whether the list is exhaustive or not.

(4b) Please precisely define what do authors mean by "overwrite" and "overrule". It is often mentioned that some processes can overwrite/overrule other processes. It is not clear what is the precise meaning of this. For example, in Figure 3, it is written that S3-4 overrules S2* but both the arrows are present in the figure.

(4c) This overruling appears a bit arbitrary. A given process can overrule many other processes. Has all such possibilities been considered?

(4d) Please also explain why certain nucleosome assembly/disassembly/sliding processes are called "regulated" or "constitutive".

Reviewer #3:

This paper offers a well-written analysis of nucleosome dynamics at the classical yeast *PHO5* promoter. It sets out a hierarchical maximum-likelihood scheme for successively fitting coarse-grained models to heterogeneous datasets, while avoiding the parametric morass, building on the work of Blasi et al., 2016. The main results (Figure 5), draw attention to patterns of fluxes between nucleosome configurations which are common among classes of the best-fitted models. Overall, this is a well-conducted study in the art of rigorous model fitting to complex data. The biological implications, however, are relegated to the Discussion. On the one hand, this presents an informed review of the experimental literature; on the other hand, the main suggestion, to reconsider the role of nucleosome assembly (Discussion), does not seem sufficiently well supported by the actual results (below). I therefore feel that the paper may be better suited to a more specialised audience in computational biology. I believe peer review is improved by transparency and therefore do not wish to remain anonymous.

(1) "We suggest to reconsider the common view for the case of the *PHO5* promoter". The models considered in the paper are coarse-grained Markov processes describing nucleosome configurations and transitions between them. In particular, neither transcription factors like *PHO4* nor nucleosome remodellers are explicitly represented. Without further analysis, it is not straightforward to draw well-founded conclusions about what may, or may not, be happening at this more detailed level. To put this in perspective, if one started from a detailed Markov process model, which included, for instance, nucleosome remodellers, and attempted to construct from that a more coarse-grained model, then the latter may not even be Markovian. It is reasonable to draw conclusions at the level of abstraction of the models themselves, as in Figure 5, but to assume that this also gives insight into what is happening at a more detailed level seems to require careful justification, which is not provided here.

(2) Surprisingly, the paper does not use the fitted models to suggest experiments which could determine the relative roles of assembly and dis-assembly at the *PHO4* promoter. This would have been a more compelling way of drawing biological significance from the modelling. Lurking behind this suggestion is the broader issue as to what should be expected from a model in biology. This is a matter on which much has been written, so I feel it is only fair to acknowledge my own bias in this respect. I believe models are not descriptions of biological reality but, rather, descriptions of our assumptions about reality (PMID 24886484). From this perspective, a model in biology does not offer predictive capability, as it does in physics or engineering but, rather, a test of its assumptions. The best test of any assumptions are data from new kinds of experiments. I was disappointed that such experiments were not suggested here.

---

## [Author Response]

Reviewer #1:[…] There is one issue which requires clarification. The likelihood ratio is used to select preferable models. However, the models compared have different number of parameters. Why not use a penalized likelihood, e.g., BIC or AIC? The underlying problem is that the large number of models has a downside: While it avoids modeling bias, it introduces an unwanted multiple testing problem.

We agree with the reviewer that if one compares models with different numbers of parameters, it is important to use model selection criteria that take this into account, ideally by using information-based criteria such as BIC and AIC. In our case, following the principle of Occam’s razor, we sought to identify only the simplest models (with fewest parameters) that are able to describe the experimental data within experimental accuracy. For this purpose, we were not forced to compare models with different parameter numbers. Instead, we started with the simplest class of on-off-slide models, which have 4 fitted parameters. None of these models were able to simultaneously describe all experimental data. We then successively increased the number of parameters, finding that 7 is the minimum number of fitted parameters needed to simultaneously describe all experimental data. Subsequently, we only needed to score these 7-parameter models relative to each other, for which the likelihood ratio provides a statistically adequate scoring scheme.

Overall, the paper is technically very convincing and the integration of the experimental data into the modeling process is excellent.Figures are clear and informative.In terms of language, the authors speak of "fit parameter". It would be better to use "fitted" to avoid possible confusion with "fit" like in "fitness".

We agree and changed the phrasing as suggested to avoid any confusion.

Reviewer #2:[…] Even though earlier studies have attempted a similar approach, this is a nice work as it is comprehensive and systematic. However, there are some concerns and some suggestions that could improve the paper.(1) Authors account only those eight promoter nucleosome configurations that are presented in the Brown et al/Small et al. experiments. However, to understand gene regulation, binding of other transcription factors (like Pho4) too are crucial. Binding of these factors are coupled with nucleosome dynamics. Can the model provide biologically relevant predictions without accounting for these factors? This is not clear from the current manuscript.

This is an important point and we apologize for not making this sufficiently clear in the manuscript. Our “regulated on-off-slide models” provide a coarse-grained description of the nucleosome dynamics, which is tailored to the available data. However, the effect of transcription factors (*Pho4* in particular) is indirectly taken into account via the “regulated” part of our models, i.e. the fact that at least one of the on-off-slide processes is regulated as a function of the promoter state. In other words, the rate constant of the regulated process(es) depends on whether the promoter is in the active or in the repressed state. This effect can be mechanistically caused by the transcription factor *Pho4* and/or other factors (such as recruited remodeling enzymes) that are not directly included in our model. Thus, even though our model is coarse-grained, we can draw biologically relevant conclusions.

In the revised version of the manuscript, we improved the explanation of our modeling rationale to clarify this important point.

Changes in subsubsection “Regulated on-off-slide models – Regulation”:

“The changing rates of regulated processes are supposed to model the effects caused by transcription factors and/or other factors which influence the promoter state, such as recruited remodeling enzymes, in a coarse-grained fashion.”

(1a) Authors find that certain sliding events are crucial. However, binding of Pho4 would provide steric hinderance and prevent many of the sliding moves. How would you reconcile this given the results presented here.

Steric hindrance by *Pho4* could reduce nucleosome sliding, but need not prevent it, since nucleosome sliding is mediated by ATP-dependent remodeling enzymes. For example, sliding by remodelers of the SWI/SNF class was shown not to be inhibited by pre-bound transcription factors, while such factors do encumber sliding by remodelers of the ISWI class (Li et al., 2015, *eLife*, PMID: 26047462). Our coarse-grained model accounts for such molecular processes via the *effective* rate constants for sliding, which subsumes the combined effect of all relevant processes.

(1b) Are there known mechanisms for directed sliding of nucleosomes? It is mentioned that the sliding speed is consistent with SWI/SNF, RSC remodelling complexes. But are any of these enzymes known to slide nucleosomes in a specific direction?

Yes, effective directional sliding has been demonstrated early on in vitro relative to DNA ends for remodelers of the ISWI and CHD class (Brehm et al., 2000, EMBO J., PMID: 10944116) and, for example, recently followed up (Levendosky and Bowman, 2019, *eLife*, PMID: 31094676). It was also shown in vivo for the ISW2 remodeling complex (Whitehouse et al., 2007) and suggested for the RSC complex in the context of poly(dA:dT) elements (Krietenstein et al., 2016). We added this information to the manuscript.

Changes in subsubsection “Regulated on-off-slide models – Site-specific sliding processes”:

“To allow directional sliding, as observed for example in vivo for ISW2 and suggested for RSC (Whitehouse et al., 2007; Krietenstein et al., 2016), we included five optional site-specific sliding processes.

(2) Even though experiments have measured positioning of only three nucleosomes, isn't it possible that the positioning of these nucleosomes is influenced by the 4th nucleosome (N-4) or even the +1 nucleosome? It is known that positioning of neighbouring nucleosomes are correlated (due to their steric exclusion). Will extending the model with the 4th nucleosome (or +1 nucleosome) provide more insights about the dynamics of the 3 observed nucleosomes? If possible, that would be a new contribution beyond all the existing models.

It was previously shown that a truncated *PHO5* promoter harboring only the N-1, N-2, and N-3 nucleosomes recapitulates the main features of the wild type *PHO5* promoter (Fascher et al., 1993). This was also a key justification for the experimental study of the Boeger group to limit their psoralen-TEM assay to these three nucleosomes. Given that our modeling approach relies on the observed configurational statistics of nucleosomes (i.e., the Boeger data), we also limited ourselves to these three nucleosomes. We felt that if we were to make assumptions about the statistics of the neighboring nucleosomes that we cannot directly derive from experimental data, this could potentially introduce a bias into our analysis, which would weaken our approach. As a consequence, we cannot distinguish effects of neighboring nucleosomes (N-4 or N+1) on the measured nucleosomes from similar effects with other causes. For example, the disappearance of the N-3 nucleosome (without any other changes regarding the N-2 and N-1 position), could be due to disassembly, but also due to sliding to the N-4 position, which we cannot distinguish. Such sliding would contribute to the disassembly processes at N-3 and is therefore subsumed in our parameters. Collectively, an analysis of all four nucleosomes of the wild type *PHO5* promoter would be interesting but no corresponding data, e.g., of the psoralen-TEM type, are currently available.

(3) Given that many studies on this topic exist, what are the new testable predictions from this study? Please write a small subsection on the new testable predictions.

We completely agree that a focused subsection on the new testable predictions, which also discusses possible experiments to test these predictions, would be a valuable addition to our manuscript. We therefore wrote precisely such a subsection, entitled “Experimentally testable predictions”, within our Discussion section, accompanied by three new supplementary figures. In addition to the one experimental test that we had already proposed in the Discussion of our originally submitted version, the new version also contains an additional subsection “Future applications of our newly established approach” featuring several new studies involving mutated *PHO5* promoters or strain backgrounds.

(4) Some suggestions to bring more clarity to the manuscript:(4a) It is good to clearly define what is a "model". In this context, typically, a master equation would define a model. When authors talk about several models, it is good to mention how precisely they define a model. Readers need to understand how did authors enumerate all the models and whether the list is exhaustive or not.

We apologize for the lack of clarity in the model definitions. The original manuscript focused on the description of the possible “processes” (assembly, disassembly and sliding) at different hierarchies (global, site-specific and configuration specific) and with or without regulation (regulated vs. constitutive). These processes are all possible building blocks that define each individual model by defining the structure of the transition rate matrix Q, with the Master equation d/dt p = p Q, p being the row vector of probabilities for the eight configurations. When fitting a different promoter state, only the transition rates of the regulated processes can change, and thus only the entries in Q which correspond to the regulated processes. The rates of constitutive processes are the same in each promoter state. In the revised manuscript, we improved the model definition and also the explanation of the combinatorial enumeration of the models.

Changes in subsection “Regulated on-off-slide models”:

“These processes, which will be explained in the following, are the possible building blocks that define each model, i.e. determine the transition rate matrix in the Master equation of the underlying Markov process for all three promoter states (Materials and methods). To describe all three promoter states within the same model, at least one of its processes has to be regulated, i.e. change its rate to achieve different configuration occurrences and dynamics between different promoter states.”

Changes in subsection “Regulated on-off-slide models – Resulting model set”:

“After taking into account all combinations of processes with up to 7 parameters in total we ended up with 68145 regulated on-off-slide models. Here we ignored duplicate models with identical transition rate matrices constructed with different processes (e.g. rare models where all reactions of a process are overruled by other processes, making it identical to the model without the overruled process).”

(4b) Please precisely define what do authors mean by "overwrite" and "overrule". It is often mentioned that some processes can overwrite/overrule other processes. It is not clear what is the precise meaning of this. For example, in Figure 3, it is written that S3-4 overrules S2* but both the arrows are present in the figure.

We apologize for this error in the legend of Figure 3. Process S3-4 does in fact not overrule S2*, as S3-4 does not govern a reaction that is part of S2*. In the example of Figure 3, only process D1-4 “overrules” the global disassembly process D, meaning that this specific disassembly reaction has a different rate, which is the fitted parameter associated with D1-4. This error in the legend of Figure 3 might have contributed to the confusion regarding the concept of overruling. We improved the manuscript by consistently using only the term “overrule” (not “overwrite”) and explaining the underlying concept more clearly.

Changes in subsubsection “Regulated on-off-slide models – Modulation by more specific processes”:

“If the reactions governed by any two processes in a given model overlap, as for example the global and any site-specific assembly process, the more specific process, i.e. the one governing less reactions, determines the rate values of these reactions, "overruling" the more general process, which then only governs the left-over, non-overlapping reactions. […] This rule is the most general solution for overlapping processes, allowing an increased as well as a decreased rate value for the reactions of the more specific process, i.e., the specific process to be enhanced or inhibited with respect to the general process.”

(4c) This overruling appears a bit arbitrary. A given process can overrule many other processes. Has all such possibilities been considered?

Yes, we considered all such possibilities. The “overruling concept” is a description of what happens in a given model where reactions governed by any two processes overlap (as it is the case with D1-4 and D). Only in this case the overruling concept applies – a specific process can only overrule less specific processes that include the same reaction. The reactions of the less specific process, which are not part of the more specific process, are left unchanged. We clarified this in the description (please refer to point 4b).

(4d) Please also explain why certain nucleosome assembly/disassembly/sliding processes are called "regulated" or "constitutive".

Please refer to our reply to point 4a.

Reviewer #3:This paper offers a well-written analysis of nucleosome dynamics at the classical yeast PHO5 promoter. It sets out a hierarchical maximum-likelihood scheme for successively fitting coarse-grained models to heterogeneous datasets, while avoiding the parametric morass, building on the work of Blasi et al., 2016. The main results (Figure 5), draw attention to patterns of fluxes between nucleosome configurations which are common among classes of the best-fitted models. Overall, this is a well-conducted study in the art of rigorous model fitting to complex data. The biological implications, however, are relegated to the Discussion. On the one hand, this presents an informed review of the experimental literature; on the other hand, the main suggestion, to reconsider the role of nucleosome assembly (Discussion), does not seem sufficiently well supported by the actual results (below). I therefore feel that the paper may be better suited to a more specialised audience in computational biology. I believe peer review is improved by transparency and therefore do not wish to remain anonymous.

We thank the reviewer for the transparent assessment of our manuscript. We were glad to hear that the reviewer appreciates the technical execution of our study. We address the remaining concerns regarding the biological implications of our results in the following point-by-point response and the revised manuscript.

(1) "We suggest to reconsider the common view for the case of the PHO5 promoter". The models considered in the paper are coarse-grained Markov processes describing nucleosome configurations and transitions between them. In particular, neither transcription factors like PHO4 nor nucleosome remodellers are explicitly represented. Without further analysis, it is not straightforward to draw well-founded conclusions about what may, or may not, be happening at this more detailed level. To put this in perspective, if one started from a detailed Markov process model, which included, for instance, nucleosome remodellers, and attempted to construct from that a more coarse-grained model, then the latter may not even be Markovian. It is reasonable to draw conclusions at the level of abstraction of the models themselves, as in Figure 5, but to assume that this also gives insight into what is happening at a more detailed level seems to require careful justification, which is not provided here.

We agree with the reviewer that an analysis of a coarse-grained model does not permit direct conclusions about molecular mechanisms. However, we would like to stress that the primary (and robust) conclusion of our analysis is directly on the same coarse-grained level as our analysis: We found that the experimental data sets were only compatible with the regulated assembly models (rather than regulated disassembly). This conclusion is directly supported by all of the seven identified models that survived the model selection process. What we wrote in the Discussion section of the originally submitted manuscript was merely a discussion of possible molecular mechanisms that would appear as regulated assembly processes on the coarse-grained level. This discussion on the molecular level also relied on a number of prior experimental results, which help in evaluating the plausibility of different possible molecular scenarios that could rationalize regulated nucleosome assembly at the *PHO5* promoter.

In the light of this comment by the reviewer, as well as point 3 of reviewer 2 and the advice of the Editor, we rewrote our Discussion to provide a better link between the direct findings of our analysis and the biological conclusions that we draw from them. At the same time, our revised manuscript provides a more thorough discussion of possible experiments to test our predictions (see also the next point) and further apply our modeling approach.

(2) Surprisingly, the paper does not use the fitted models to suggest experiments which could determine the relative roles of assembly and dis-assembly at the PHO4 promoter. This would have been a more compelling way of drawing biological significance from the modelling. Lurking behind this suggestion is the broader issue as to what should be expected from a model in biology. This is a matter on which much has been written, so I feel it is only fair to acknowledge my own bias in this respect. I believe models are not descriptions of biological reality but, rather, descriptions of our assumptions about reality (PMID 24886484). From this perspective, a model in biology does not offer predictive capability, as it does in physics or engineering but, rather, a test of its assumptions. The best test of any assumptions are data from new kinds of experiments. I was disappointed that such experiments were not suggested here.

We added the new section “Experimentally testable predictions” to the Discussion, where we describe testable predictions of Flag/Myc dynamics at the *PHO5* promoter as well as chromatin opening and closing experiments to investigate the dynamics of nucleosome configurations.

Additionally in the new section “Future applications of our newly established approach”, we describe experiments to investigate the nucleosome dynamics upon several promoter/strain mutations or other experimental manipulations that might give further insights, like *Pho4* binding site deletions, remodeler enzyme deletions and assays with cell cycle arrested cells.